# Microtubules oppose cortical actomyosin-driven membrane ingression during *C. elegans* meiosis I polar body extrusion

**Alyssa R. Quiogue**[1], **Eisuke Sumiyoshi**[1], **Adam Fries**[1,2], **Chien-Hui Chuang**[1], **Bruce Bowerman**[1]*

**1** Institute of Molecular Biology, University of Oregon, Eugen, Oregon, United States of America, **2** Imaging Core, Office of the Vice President for Research University of Oregon, Eugene, Oregon, United States of America

* bowerman@uoregon.edu

**Data Availability Statement:** All relevant data are within the manuscript and its Supporting Information files.

## Abstract

During *C. elegans* oocyte meiosis I cytokinesis and polar body extrusion, cortical actomyosin is locally remodeled to assemble a contractile ring that forms within and remains part of a much larger and actively contractile cortical actomyosin network. This network both mediates contractile ring dynamics and generates shallow ingressions throughout the oocyte cortex during polar body extrusion. Based on our analysis of requirements for CLS-2, a member of the CLASP family of proteins that stabilize microtubules, we recently proposed that a balance of actomyosin-mediated tension and microtubule-mediated stiffness limits membrane ingression throughout the oocyte during meiosis I polar body extrusion. Here, using live cell imaging and fluorescent protein fusions, we show that CLS-2 is part of a group of kinetochore proteins, including the scaffold KNL-1 and the kinase BUB-1, that also co-localize during meiosis I to structures called linear elements, which are present within the assembling oocyte spindle and also are distributed throughout the oocyte in proximity to, but appearing to underlie, the actomyosin cortex. We further show that KNL-1 and BUB-1, like CLS-2, promote the proper organization of sub-cortical microtubules and also limit membrane ingression throughout the oocyte. Moreover, nocodazole or taxol treatment to destabilize or stabilize oocyte microtubules leads to, respectively, excess or decreased membrane ingression throughout the oocyte. Furthermore, taxol treatment, and genetic backgrounds that elevate the levels of cortically associated microtubules, both suppress excess membrane ingression in *cls-2* mutant oocytes. We propose that linear elements influence the organization of sub-cortical microtubules to generate a stiffness that limits cortical actomyosin-driven membrane ingression throughout the oocyte during meiosis I polar body extrusion. We discuss the possibility that this regulation of sub-cortical microtubule dynamics facilitates actomyosin contractile ring dynamics during *C. elegans* oocyte meiosis I cell division.

**Funding:** This work was funded by NIH grant R35 GM131749 (A.R.Q., E.S., C.-H.C., B.B.), NIH Training Grant T32HD007348-32 (A.R.Q.), and the University of Oregon Office of the Vice-President for Research (A.F.). The funders had no role in study design, data collection and analysis, decision to publish, or preparation of the manuscript.

**Competing interests:** The authors have declared that no competing interests exist.

## Author summary

Animal oocytes undergo a sequence of two specialized cell divisions called meiosis I and II in which three quarters of the genome content is expelled into two small, extracellular, membrane-bound compartments called polar bodies. These two meiotic divisions reduce the genome content of the oocyte from a diploid state, with two copies of each chromosome, to a haploid state, with a single copy of each chromosome. As a result, fertilization by a sperm restores the normal diploid genome content. We provide evidence that tubular cytoskeletal polymers called microtubules are enriched near the inner surface the of oocyte cell membrane, where they provide a stiffness that resists membrane ingressions driven by the force-generating cortical actomyosin cytoskeleton. These membrane ingressions occur throughout the oocyte during meiotic polar body extrusion. Our results suggest that a balance of microtubule-mediated stiffness and actomyosin-mediated membrane ingression throughout the oocyte may be important for the proper assembly and function of the contractile ring that pinches off polar bodies at one end of the oocyte during oocyte meiotic cell division.

## Introduction

Animal cell shape and morphogenesis are influenced by both tension and elasticity within the cell cortex [1–4]. Cortical tension and elasticity both depend on the actomyosin cytoskeleton and its associated proteins. Non-muscle myosin and microfilament architecture are largely responsible for generating tension [5], while increased cross-linking of cortical microfilaments to the plasma membrane by Ezrin/Radixin/Moesin (ERM) proteins during the cell rounding associated with mitosis generates decreased elasticity and hence increased cortical stiffness [6–8].

While there is growing evidence for crosstalk between the microtubule and microfilament cytoskeletons [9,10], microtubules are generally viewed as forming distinct cytoskeletal structures that are not part of the animal cell cortex. Nevertheless, we recently proposed that cortically associated microtubules stiffen the *C. elegans* oocyte cortex to limit actomyosin-driven membrane ingression throughout the oocyte surface during the extrusion of discarded chromosomes into a small polar body [11]. This hypothesis followed from investigating the requirements during oocyte meiosis I cell division for CLS-2, a *C. elegans* member of the widely conserved CLASP family of proteins that bind to microtubule plus ends and promote stability through TOG domains that bind tubulins [12–14].

During oocyte meiotic cell division, CLS-2 localizes to spindle microtubules, to kinetochores, and to small patches called linear elements or rods that are associated with the assembling spindle and also distributed throughout the oocyte in proximity to its cortex [15–17]. We previously reported two prominent defects during meiosis I cell division in mutant oocytes produced by worms homozygous for null mutations in the *cls-2* locus [11]. First, spindle microtubule levels were reduced, and spindles never become bipolar, with chromosomes failing to separate or be extruded into a polar body. Second, abnormally deep membrane ingressions occurred throughout the entire oocyte, even though the cortical actomyosin network appeared normal. Thus CLS-2 has at least two spatially distinct functions during oocyte meiosis I cell division: (i) promoting spindle microtubule stability and bipolar spindle assembly, and (ii) limiting the extent of membrane ingression throughout the oocyte during polar body extrusion.

While the requirements for cortical actomyosin during meiotic cell division and polar body extrusion in *C. elegans* have been investigated more extensively [18–22], microtubules also are present throughout the oocyte in proximity to the cortex during meiosis I [11,15,23–25]. However, the functional importance of these cortically associated oocyte microtubules remains unclear. Because CLS-2 orthologs are known to promote microtubule stability, and no differences in cortical actomyosin dynamics were detected in *cls-2* mutants compared to control oocytes [11], we hypothesized that CLS-2 within linear elements distributed throughout the oocyte stabilizes cortically associated microtubules to stiffen the cortex and oppose the actomyosin-driven membrane ingressions that occur throughout the oocyte during meiosis I polar body extrusion (Fig 1). This regulation may be important for proper assembly and ingression of the small, spindle-associated meiosis I contractile ring, which forms as part of the much larger cortical actomyosin network (Fig 1; see Discussion).

Other observations also suggest that microtubules may influence cortical function in *C. elegans* oocytes. In addition to CLS-2, two other widely conserved proteins are known to regulate microtubules both within the meiotic spindle and throughout the oocyte: the kinesin 13/MCAK microtubule depolymerase family member called KLP-7, and another TOG domain protein, ZYG-9/chTOG. Both KLP-7 and ZYG-9 are required for oocyte meiotic spindle assembly, and reducing the function of either results in elevated levels of both spindle microtubules and microtubules enriched near the cortex throughout the oocyte [24–26]. While KLP-7 and ZYG-9 clearly regulate their levels, the functional significance of these cortically associated microtubules and their regulation by KLP-7 and ZYG-9 have received little attention.

In addition to its roles in oocyte meiotic spindle assembly and polar body extrusion, CLS-2 also acts early in mitosis to stabilize microtubule/kinetochore attachments [27–29]. In both meiotic oocytes and mitotic blastomeres, CLS-2 is part of an outer kinetochore sub-complex that includes the spindle assembly checkpoint kinase BUB-1 and the redundant coiled-coil proteins and CENP-F orthologs HCP-1 and -2. Like other outer kinetochore sub-complexes, the BUB-1/HCP-1/2/CLS-2 sub-complex depends on the scaffolding protein KNL-1 for kinetochore localization [16,27]. Like CLS-2, both KNL-1 and BUB-1 also localize to linear elements that are enriched within the assembling spindle and also distributed throughout the oocyte during most of meiosis I cell division [15–17]. While all three proteins co-localize to kinetochores, it is not known whether they also co-localize within the linear elements.

The spindle-associated linear elements are thought to promote kinetochore expansion and hence the capture of spindle microtubules during mitotic cell division, with similar structures present in mammalian cells [30]. The role(s) played by the more widely distributed linear elements observed away from the spindle during oocyte meiosis I are not known. Here we report our investigation of the requirements for KNL-1, BUB-1 and CLS-2 during *C. elegans* polar body extrusion. Our results support a model in which linear elements distributed throughout the oocyte in proximity to the actomyosin cortex influence the organization of microtubules to stiffen the sub-cortex and thereby limit the extent of cortical actomyosin-driven membrane ingression during *C. elegans* meiosis I polar body extrusion.

## Results

### KNL-1, BUB-1 and CLS-2 co-localize to linear elements that underlie the cortex during meiosis I

To determine when KNL-1, BUB-1 and CLS-2 are present, and if they co-localize to linear elements, we used spinning disk confocal microscopy and live cell imaging to track fluorescent protein fusions to each protein (see Materials and Methods). First, we examined their localization throughout meiosis I and II, imaging *in utero* oocytes that express GFP fused to one of the

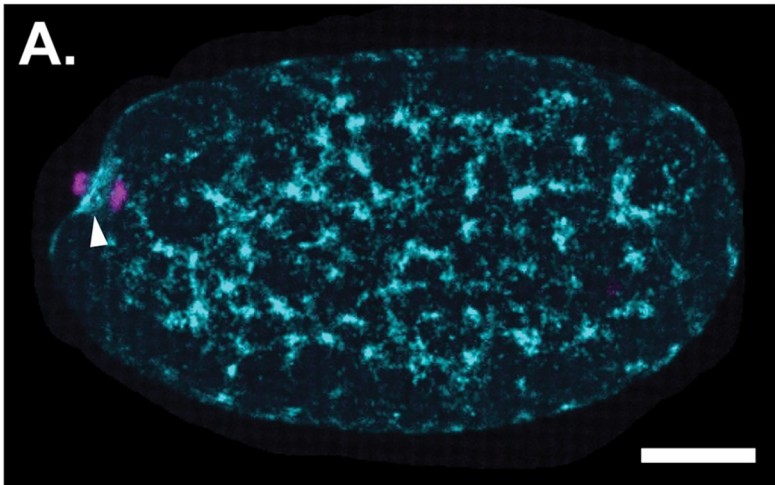

GFP::NMY-2;mCherry::H2B

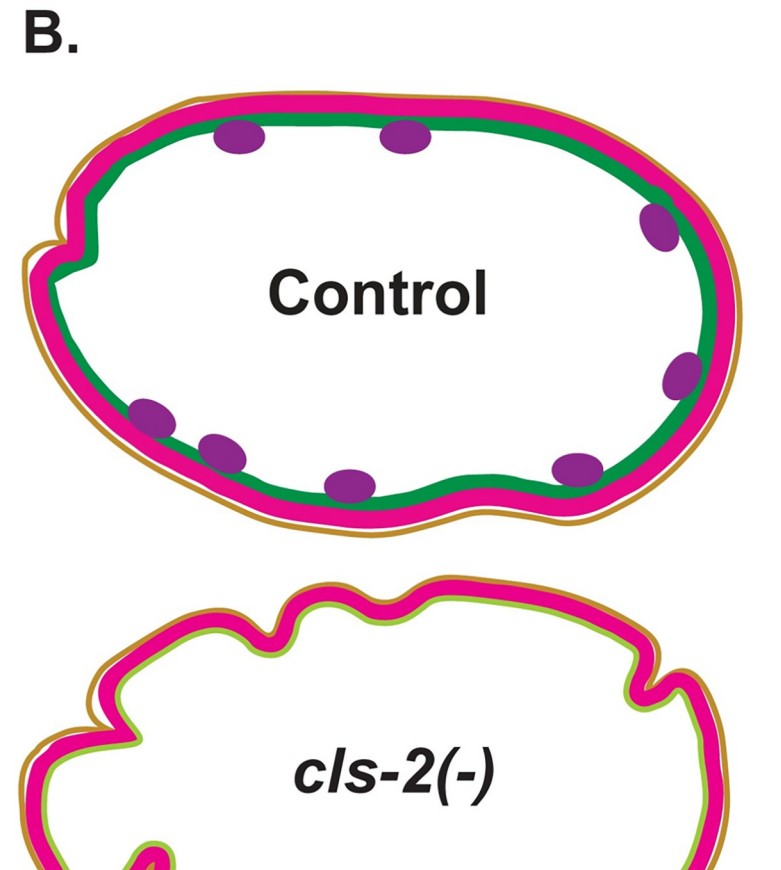

oocyte membrane actomyosin contractility
microtubule stiffness CLS-2 patches

**Fig 1. Model in which microtubules stiffen the sub-cortex to limit cortical actomyosin-driven membrane ingression throughout the oocyte during *C. elegans* meiosis I polar body extrusion.** (A) Merged maximum intensity projections (MIPs) during meiosis I anaphase of a live *ex utero* oocyte expressing a GFP fusion to the nonmuscle myosin II NMY-2 (cyan) and an mCherry fusion to a histone (magenta) to mark the actomyosin cortex and chromosomes, respectively. MIP of five surface-most planes showing cortical non-muscle myosin II is merged with MIP of five consecutive focal planes that encompass most of the oocyte chromosomes, visible at the left, anterior end of the oocyte. During polar body extrusion, the cortical actomyosin network includes a contractile ring (arrowhead) that assembles near the oocyte chromosomes and ultimately constricts between separating chromosomes. (B, C) Model with schematics that depict (B) CLS-2 linear elements (purple) promoting cortically associated microtubule-mediated stiffness (green) that limits cortical actomyosin-driven (red) membrane ingressions (tan) throughout a control oocyte; and (C) a *cls-2* null mutant oocyte with increased membrane ingression due to reduced sub-cortical stiffness caused by altered organization of cortically associated microtubules, with an unchanged level of cortical actomyosin contractility. Scale bar = 10 μm.

three kinetochore proteins, and mCherry fused to a histone (mCherry::H2B) to mark chromosomes. As reported previously [11,15,16], we detected the GFP fusions to KNL-1, BUB-1 and CLS-2 shortly after nuclear envelope breakdown during meiosis I, in association with chromosomes and also in linear elements enriched within the assembling spindle and present less densely throughout the oocyte (Figs 2A and S1 and S1 Movie). The KNL-1, BUB-1, and CLS-2 linear elements persisted until the beginning of anaphase B, when they became undetectable, approximately 13 minutes after nuclear envelope breakdown and just as polar body extrusion began. These linear elements varied in size and shape and were mobile, sometimes fusing together or fragmenting into pieces both near the spindle and throughout the oocyte (S1 Movie), and they generally became less linear and more patch-like as meiosis I proceeded (Figs 2A, 2B and S1 and S1 Movie).

To assess the location of the widely dispersed linear elements relative to the oocyte surface, we compared different focal planes in three-dimensional imaging data from *ex utero* oocytes expressing the different GFP fusions (Fig 2B). For all three, most of the linear elements were present within maximum intensity projections (MIPs) of the five surface-most focal planes, and they were distributed throughout the oocyte within these focal planes. By contrast, within MIPs of the five central focal planes, and in single central focal planes, the linear elements were found exclusively at the edges in proximity to the oocyte surface. While our resolution is limited, many of the linear elements detected in the central focal planes appeared to be near but not immediately adjacent to the oocyte surface, suggesting they may underlie, rather than be part of the oocyte cortex. However, in some cases, patches were adjacent to the oocyte surface. Hereafter, we refer to these cortically associated structures as sub-cortical patches, to distinguish them from the spindle associated linear elements.

Subsequent to anaphase B of meiosis I, as documented previously [15,16], KNL-1, BUB-1 and CLS-2 exhibited overlapping but distinct dynamics. After the KNL-1, BUB-1 and CLS-2 sub-cortical patches dissipated during anaphase A of meiosis I, we subsequently detected BUB-1 and CLS-2, but not KNL-1 at the central spindle between the separating chromosomes. During meiosis II, we again detected all three in association with chromosomes, and BUB-1 and CLS-2 at the central spindle, but no linear elements or sub-cortical patches were detected near the spindle or throughout the oocyte (Figs 2A and S1 and S1 Movie).

We next asked if KNL-1, BUB-1 and CLS-2 co-localize to the sub-cortical patches during meiosis I, in oocytes expressing mCherry::H2B to mark chromosomes and different combinations of RFP and GFP fusions to KNL-1, BUB-1, and CLS-2. Using live imaging of *ex utero* oocytes, we saw that both BUB-1::GFP and GFP::CLS-2 co-localized with RFP::KNL-1, and BUB-1::RFP co-localized with GFP::CLS-2, at most if not all sub-cortical patches during meiosis I (Fig 3A and S2 Movie).

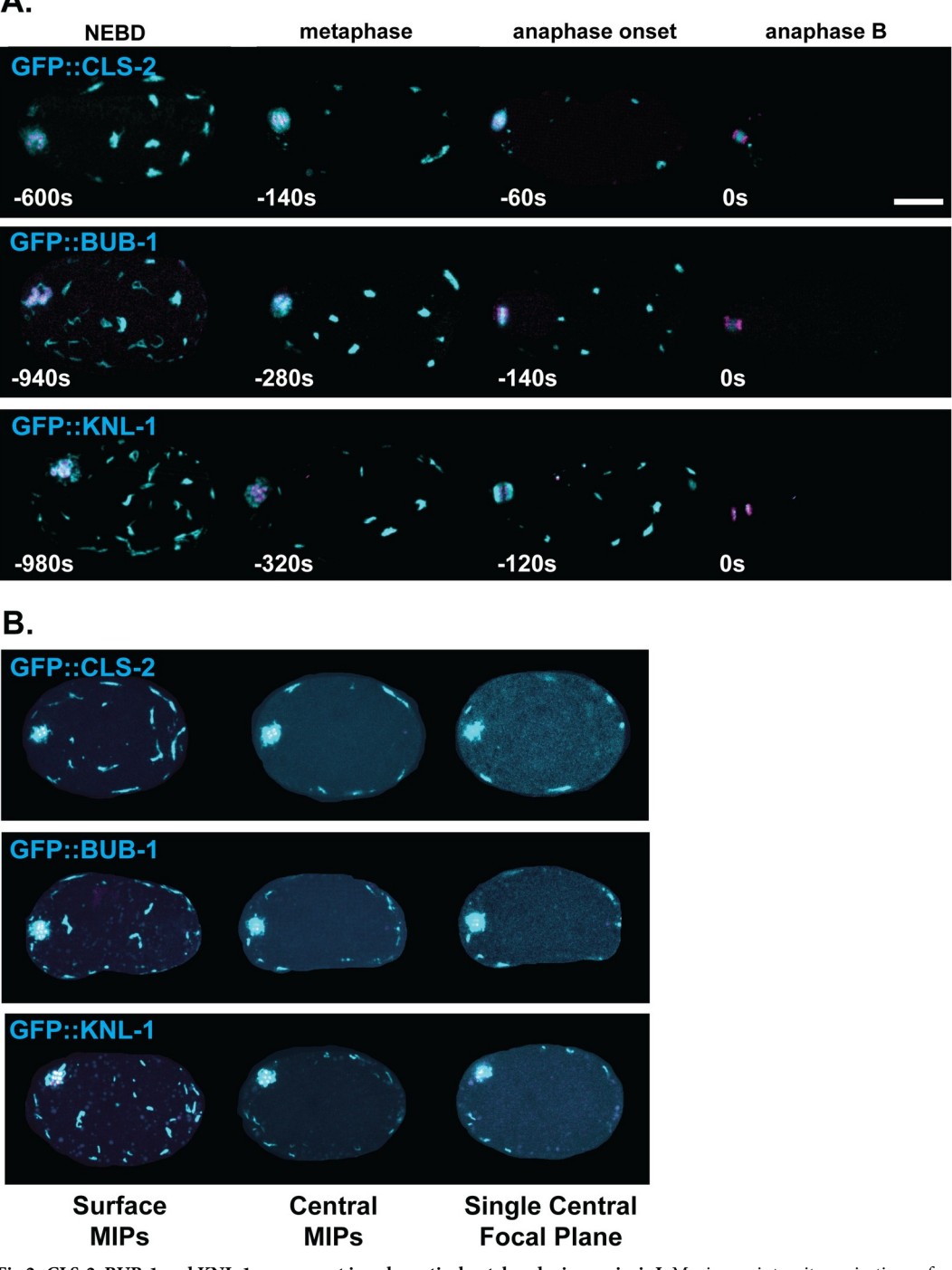

**Fig 2. CLS-2, BUB-1 and KNL-1 are present in sub-cortical patches during meiosis I.** Maximum intensity projections of five surface-most focal planes, merged with five internal focal planes that include most of the chromosomes, from twenty focal planes in live *in utero* oocytes (A), and from 16 focal planes in live *ex utero* oocytes (B), collected in 1 μm steps during meiosis I in oocytes expressing mCherry::H2B to mark chromosomes (magenta) and GFP fusions to CLS-2, BUB-1, and KNL-1 (cyan). The background was enhanced in (B) to highlight oocyte boundaries. All three kinetochore proteins localized to chromosomes (visible at left, anterior end of the oocytes) and to linear elements within the oocyte spindle and also distributed throughout the oocyte near the cortex, beginning at nuclear envelope breakdown (NEBD). The linear elements became more patch-like over time and persisted until the beginning of anaphase B, when they were no longer detectable. In this and all subsequent figures, unless otherwise noted, time is in seconds, with t = 0 denoting the beginning of anaphase B, when the chromosomes were most condensed (see Materials and Methods). Scale bar = 10 μm.

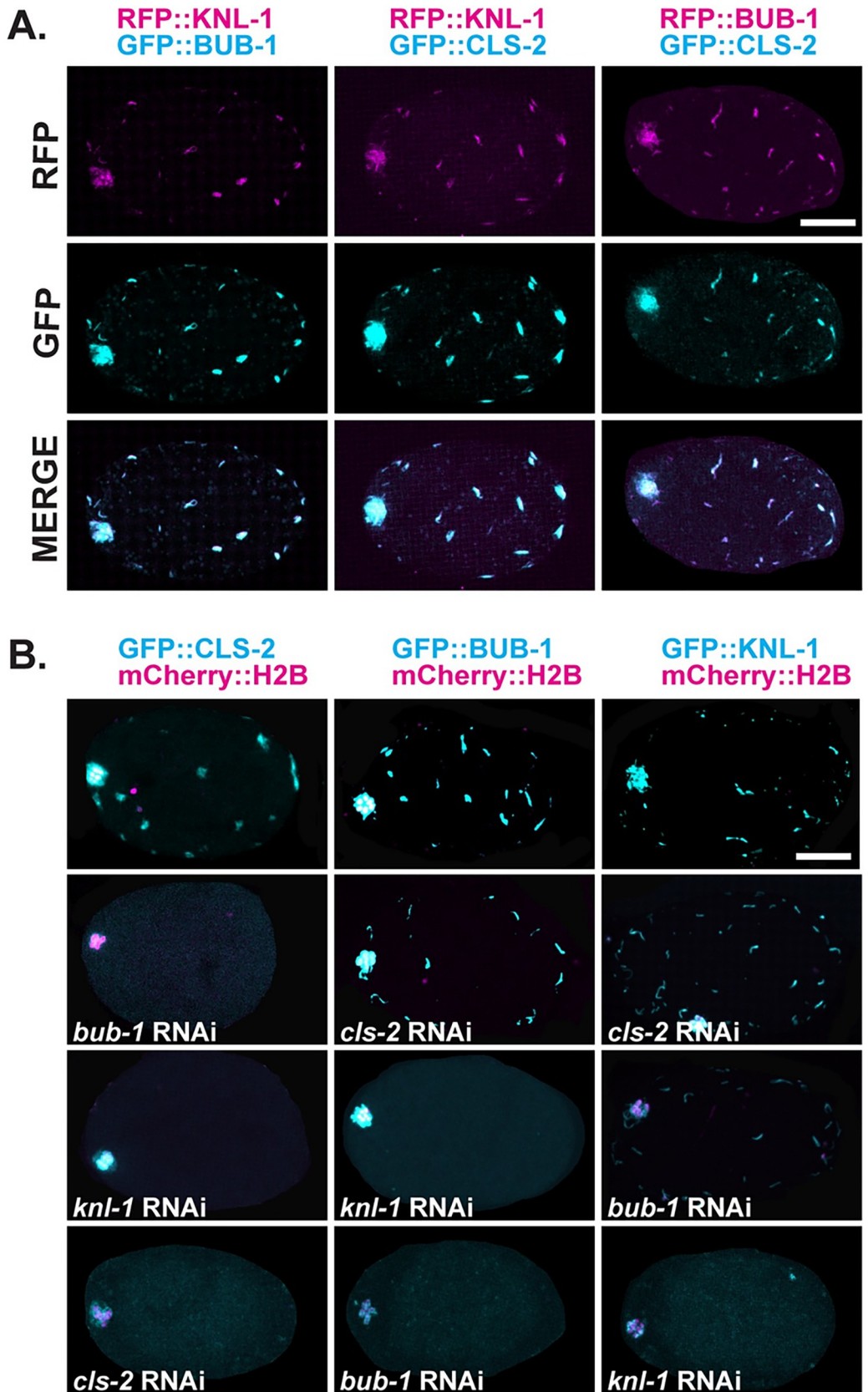

**Fig 3. KNL-1, BUB-1 and CLS-2 exhibit some mutual dependency for localization to sub-cortical patches.** (A) Maximum intensity projections (MIPs) of five surface-most focal planes during meiosis I in *ex utero* oocytes expressing both RFP (magenta) and GFP (cyan) fusions as follows: RFP::KNL-1 and BUB-1::GFP (left column), RFP::KNL-1 and GFP::CLS-2 (middle column), or RFP::BUB-1 and GFP::CLS-2 (right column). All images are from *ex utero* oocytes during metaphase, shortly before anaphase chromosome movement, when the patches were numerous and bright. Each pair of proteins co-localized to all patches. Spindle-associated GFP signal is visible at the left, anterior end of each oocyte. (B) Merged MIPs of five surface-most focal planes, and five consecutive internal focal planes that encompass most of the oocyte chromosomes, during meiosis I in *ex utero* oocytes that express mCherry::H2B (magenta) to mark chromosomes, and GFP (cyan) fusions to CLS-2, BUB-1 or KNL-1. As shown at bottom, RNAi knockdowns targeting CLS-2, BUB-1 and KNL-1 only partially reduced protein levels, with some signal remaining associated with chromosomes. See text for a summary of these results, S2 Fig for quantification of patch signal intensities, and S1 File for raw data used to calculate P values. Scale bars = 10 μm.

At oocyte kinetochores, CLS-2 requires both KNL-1 and BUB-1 for its localization, while BUB-1 requires only KNL-1, and KNL-1 does not depend on either BUB-1 or CLS-2 [16].To assess their dependencies at sub-cortical patches, we used RNA interference (RNAi) to knock down each component in oocytes expressing GFP fusions to one of the three proteins, to assess both the effectiveness of our RNAi knockdowns and the requirements for sub-cortical patch localization (see Materials and Methods). Our RNAi knockdowns did not completely reduce the levels of the proteins targeted for depletion, with signal still detectable in association with chromosomes (Fig 3B). However, when we quantified the signal detected in sub-cortical patches throughout the five surface-most focal planes during five consecutive time points prior to the start of anaphase B, we found that each RNAi knockdown nearly or completely eliminated patch signal intensity for the targeted protein (S2 Fig). We then found that CLS-2 was undetectable in cortical patches after knocking down either KNL-1 or BUB-1 (Figs 3B and S2), consistent with their relationships at kinetochores. For BUB-1, about 90% of the BUB-1::GFP patch signal was lost after KNL-1 knockdown, much as has been observed at kinetochores, with the residual BUB-1 perhaps being due to incomplete knockdown. However, we also saw a roughly 50% reduction in BUB-1 patch signal after CLS-2 knockdown (Figs 3B and S2), whereas as BUB-1 has been reported to localize to kinetochores independently of CLS-2 [16]. Finally, while KNL-1 was reported to localize to kinetochores independently of both BUB-1 and CLS-2 [16], we found that KNL-1 patch levels were reduced by nearly 50% after BUB-1 knockdown, and we saw some reduction in KNL-1 patch signal after CLS-2 knockdown (Figs 3B and S2). To summarize, during *C. elegans* oocyte meiosis I cell division, the kinetochore proteins KNL-1, BUB-1 and CLS-2 co-localize to dynamic sub-cortical patches that appear upon nuclear envelope breakdown and persist until the beginning of anaphase B during meiosis I, with greater mutual dependence for patch localization than has been reported for kinetochores.

## KNL-1, BUB-1 and CLS-2 are each required to limit membrane ingression during meiosis I polar body extrusion

To determine whether BUB-1 and KNL-1, like CLS-2 [11], are required to limit membrane ingression throughout the oocyte during meiosis I polar body extrusion, we examined membrane dynamics in control and in *cls-2(or1948)* null mutants, and after using RNAi to knockdown KNL-1 or BUB-1, in oocytes that express GFP fused to a plasma membrane marker (GFP::PH) and the chromosome marker mCherry::H2B (see Materials and Methods). Because chromosome separation fails in *cls-2* mutant oocytes [11,14,16], we defined the beginning of anaphase B as the timepoint when the oocyte chromosome mass was most condensed [31], for all control and mutant oocytes. The GFP::PH fusion encodes a rat phospholipase C PH domain that is known to bind PIP2 membrane lipids [32]. While this domain might interfere

with cortical dynamics, the membrane ingressions observed in control and mutant oocytes resemble those observed using Nomarski optics in oocytes that lack the GFP::PH fusion. In control oocytes, spindle associated furrows began to ingress near the beginning of anaphase B and constricted as meiosis progressed. The membrane ingressions associated with polar body extrusion were accompanied by multiple shallow ingressions throughout the oocyte during anaphase B (Figs 4A and S3 and S3 Movie). During polar body extrusion attempts in *cls-2* mutant oocytes, abnormally deep membrane ingressions appeared throughout the oocyte, as previously reported [11], with the extent of ingression peaking late in anaphase B and then rapidly declining (Figs 4A, 4B and S3 and S3 Movie). After RNAi knockdown of either KNL-1 or BUB-1, we also observed abnormally deep membrane ingressions that were variably patterned around the oocyte and indistinguishable from those observed in *cls-2* mutants (Figs 4C, 4D and S3 and S3 Movie).

To quantify membrane ingression during polar body extrusion, we developed a data analysis pipeline to objectively measure the number and length of all ingressions during anaphase B (S4C and S4D Fig). In brief, we used a convex hull of the oocyte contour, from a single central focal plane, as a reference for measuring furrow appearance and length, and we wrote a program that quantifies the number and length of all furrows. We manually excluded furrows associated with the oocyte chromosomes, filtered all furrows less than 1 μm in length, and quantified ingression length and number over a normalized anaphase B timescale. The mean length of ingressions, and the mean sum of lengths, were significantly increased in *knl-1*, *bub-1* and *cls-2* mutants compared to control oocytes (Fig 4E–4G). We also detected a significant increase in the number of ingressions 1 μm or greater in length after both KNL-1 and BUB-1 knockdowns, but not in *cls-2 (or1948)* oocytes (Fig 4E–4G), even though the RNAi knockdowns only partially reduce gene function (see Figs 3B and S2) while the *or1948* allele eliminates all *cls-2* function. This difference suggests that KNL-1/BUB-1-dependent factors other than CLS-2 contribute to the negative regulation of membrane ingression during polar body extrusion (see Discussion).

In addition to KNL-1, BUB-1 and CLS-2, many other kinetochore proteins have been detected during both meiotic and mitotic cell division in structures referred to variously as rods or linear elements [15–17,30,33]. Moreover, the kinetochore and linear element component ROD-1 has been shown to form filaments in vitro, and greatly extended linear structures when over-expressed in vivo, that are thought to mediate the formation of linear elements [30]. To ask if the KNL-1/BUB-1/CLS-2 sub-cortical patches are dependent on ROD-1, we used RNAi to knock down ROD-1 in oocytes expressing mCherry::H2B to mark chromosomes and GFP fusions to KNL-1, BUB-1 and CLS-2, after verifying that our RNAi conditions significantly reduced the levels of a GFP fusion to ROD-1 (S5D Fig). We found that all three patch components were greatly reduced after knocking down ROD-1 (S2B and S5A Figs). We then used RNAi to knock down ROD-1 in oocytes expressing mCherry::H2B and the GFP::PH membrane marker, and observed increased membrane ingression throughout the oocyte during polar body extrusion (S5B and S5C Fig). We conclude that the sub-cortical patches are similar in composition to the previously described linear elements that are associated with spindles. While we cannot exclude other pools of these proteins being responsible (see Discussion), the sub-cortical patches are well positioned to regulate microtubules that limit membrane ingression throughout the oocyte during meiosis I polar body extrusion.

## Failures in polar body extrusion do not correlate with increased membrane ingression

We next asked if the increased membrane ingressions observed throughout oocytes after knocking down sub-cortical patch components are associated with failures in polar body

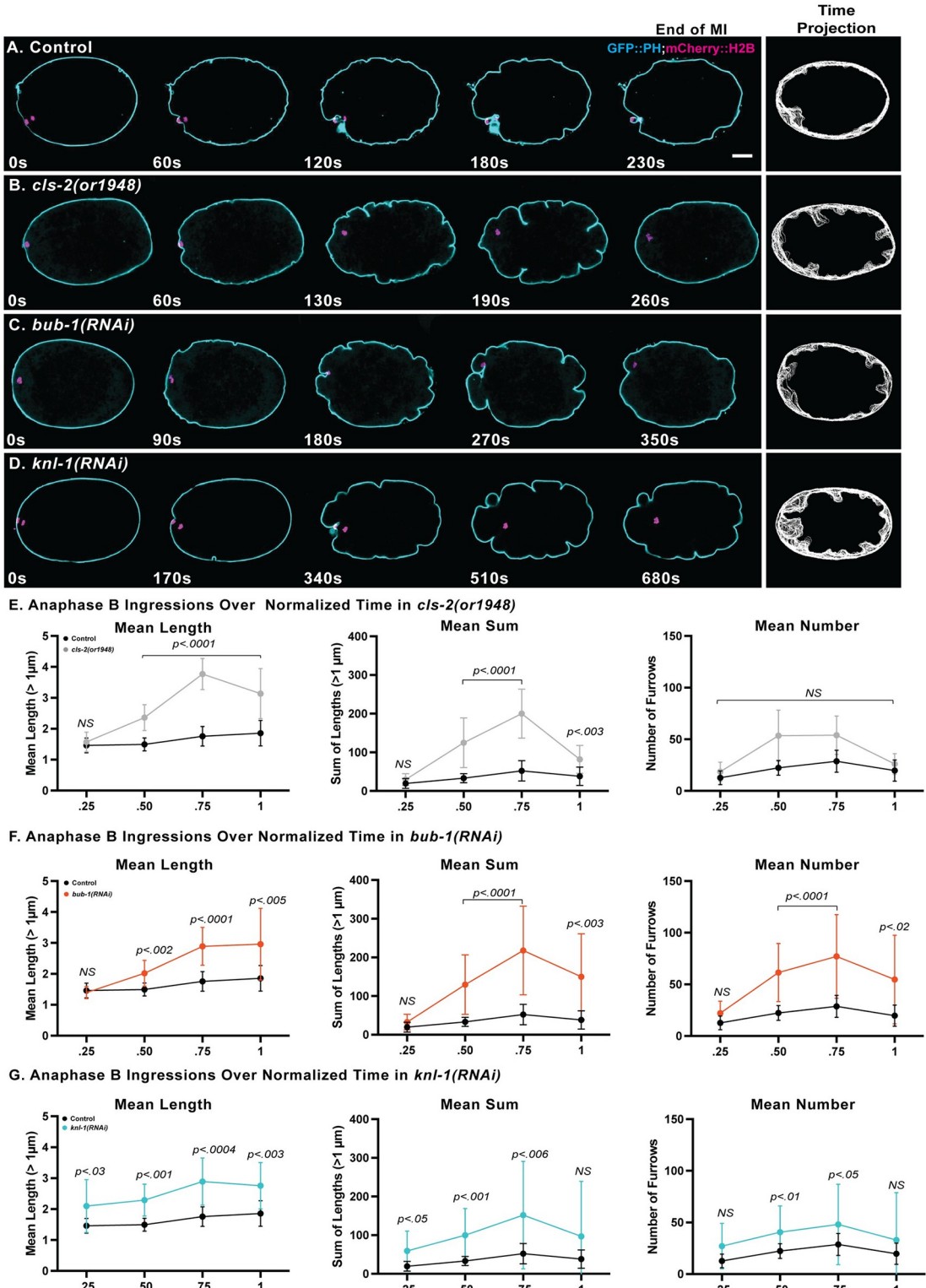

**Fig 4. KNL-1, BUB-1 and CLS-2 each limit membrane ingression during meiosis I anaphase B.** (A-D) Selected and merged focal planes from *ex utero* oocytes that express GFP::PH (cyan) and mCherry::H2B (magenta) to mark the plasma membrane and chromosomes, respectively. A single central focal plane is shown for the membrane, merged with a maximum intensity projection of five consecutive internal focal planes that encompass most of the oocyte chromosomes. Time is in seconds, with t = 0 denoting the beginning of anaphase B, when chromosomes were most condensed, and the last time point is the end of meiosis I, when

oocyte chromosomes first began to decondense prior to meiosis II (see Materials and Methods). Time projections of all central focal plane time points during anaphase B are shown to the right for each genotype. See S3 Fig for additional time projections of each genotype. Limited ingressions occurred throughout anaphase B in control oocytes (A), with excess furrowing observed in *cls-2(or1948)* null mutant oocytes (B) and after RNAi depletion of BUB-1 (C) and KNL-1 (D). (E-F) Quantification of anaphase B membrane ingressions over four equally spaced normalized anaphase B time intervals, to control for variation in cell cycle time (S3A Fig). Graphs depict the mean length of all membrane ingressions per oocyte that were 1μm or greater in length (left), the mean sum of all ingressions 1μm or greater in length (middle), and the mean number of ingressions 1μm or greater in length (right), at each time point (see Materials and Methods). The mean length, and the mean sum of lengths, both peaked late in anaphase B for all genotypes. P values denote comparisons between mutant and controls at indicated time points. For all figures, the Mann-Whitney U-test was used to calculate P-values (S2–S6 Files). See S2 File for raw data used to calculate P values for membrane ingression. Error bars and values are mean ± SEM. Scale bar = 10 μm.

extrusion. We scored polar body extrusion as successful if we could detect any mCherry::H2B signal in an external polar body after the completion of meiosis I. As reported previously [11], polar body extrusion during meiosis I failed in ~2/3 of *cls-2(or1948)* oocytes (S4B Fig). However, after RNAi knockdown of either KNL-1 or BUB-1, polar body extrusion was usually successful, with some chromosome signal detected within a polar body after the completion of meiosis I in over 80% of the mutant oocytes (S4B Fig). As our RNAi knockdowns only partially reduced these protein levels (Figs 3B and S2B), we used CRISPR/Cas9 to degron tag the endogenous *knl-1* and *bub-1* loci, and Auxin-induced degradation (AID), in an effort to more fully reduce their function during meiosis I [34] (see Materials and Methods). We found more severe defects in polar body extrusion after BUB-1 AID, with almost 90% of the oocytes failing to extrude a polar body (S4B Fig). However, we still detected a polar body in 75% of the oocytes after KNL-1 AID, and after ROD-1 RNAi we detected a polar body in 90% of the oocytes. Thus, there is little correlation between increased membrane ingression and polar body extrusion failure. However, our criterion for failed polar body extrusion may underscore the defects, and we have not extensively examined the dynamics of contractile ring assembly and ingression in these mutant backgrounds (see Discussion).

## KNL-1, BUB-1 and CLS-2 influence the organization of sub-cortical microtubules during meiosis I polar body extrusion

We next asked how KNL-1, BUB-1 and CLS-2 limit the extent of oocyte membrane ingression during meiosis I. We previously hypothesized that CLS-2 stabilizes microtubules to promote cortical stiffness and thereby limit actomyosin-driven membrane ingression during meiosis I polar body extrusion [11] (Fig 1). We also have previously shown that microtubules distributed throughout the oocyte, outside of the meiotic spindle, are enriched near the oocyte surface [25], although we did not compare the distribution of these microtubules to the distribution of actomyosin to better assess their location relative to the oocyte cortex. Furthermore, microtubule levels near the cortex throughout the oocyte are much lower than those associated with the meiotic spindle, and we were unable in our previous study to detect any significant change in cortically associated microtubule levels in *cls-2* mutant oocytes, using relatively low resolution live imaging data from *in utero* oocytes within immobilized whole mount worms [11].

To better assess whether KNL-1, BUB-1 and CLS-2 influence cortically associated microtubules during oocyte meiotic cell division, and to more closely examine the location of these microtubules relative to the oocyte cortex, we have now used higher resolution live imaging of *ex utero* oocytes that express GFP fused to either a β-tubulin (GFP::TBB-2) to mark microtubules, or the non-muscle myosin II NMY-2 (GFP::NMY-2) to mark cortical actomyosin, and mCherry::H2B to mark chromosomes. In control oocytes, we observed a network of microtubule foci distributed throughout the surface-most focal planes, whereas within the central focal planes we observed a modest enrichment of microtubule signal near the oocyte surface that

extended a few microns into the cytoplasm throughout the oocyte (Fig 5 and S4 Movie). By contrast, we found that all GFP::NMY-2 foci were closely apposed to the oocyte surface (Fig 5), as expected for cortical actomyosin [1,4]. How intimately these cortically associated microtubules intermingle with cortical actomyosin, whether and how cortical actomyosin and these cortically associated microtubules are attached to the membrane or to each other, and the extent to which microtubules are present throughout the entire volume of the oocyte, requires further investigation (see Discussion). Nevertheless, we conclude that oocyte microtubules are enriched in a layer that appears to overlap with but largely underlies the cell cortex throughout the oocyte during meiosis I polar body extrusion. Hereafter, we refer to these cortically associated microtubules as sub-cortical.

To determine whether the loss of KNL-1, BUB-1 or CLS-2 might influence the sub-cortical microtubules, we used RNAi to knock down each patch component in *ex utero* oocytes expressing GFP::TBB-2 and mCherry::H2B. In brief, for each knockdown we observed a subtle but apparent reduction in the brightness of the sub-cortical microtubule foci scattered throughout the surface-most focal planes (Fig 6A–6D and S4 Movie). Furthermore, enrichment of the microtubule signal near the oocyte surface in central focal planes was lost (Fig 6E).

We previously reported that we could not detect a significant difference in the levels of cortically associated microtubules in *cls-2* null mutant oocytes imaged *in utero*, based on a comparison of the integrated pixel intensities from mutant and control oocytes [11]. Despite the subtly altered distribution of sub-cortical microtubules in our imaging data from *ex utero* oocytes reported here (Fig 6), we nevertheless have not detected a significant difference in sub-cortical microtubule levels in mutant versus control oocytes. However, based on our perception of brighter small foci of sub-cortical microtubule signal in control oocytes, we assessed the relative standard deviation of cortically associated microtubule signal intensity within the surface-most focal planes across entire mutant and control oocytes at the beginning of anaphase B. We found a significant decrease in the relative standard deviation in *cls-2* mutants compared to control oocytes, consistent with less variance in signal intensity across the mutant oocytes during attempts at polar body extrusion (S4E Fig). We conclude that loss of CLS-2 likely leads to an alteration in microtubule organization, rather than an overall decrease in microtubule levels.

To further assess the organization of sub-cortical microtubules in control and mutant oocytes, we used Imaris software to define and count sub-Cortical Microtubule Foci, or sCMFs, during polar body extrusion (see Materials and Methods). In brief, sCMFs were defined based on the local variance of pixel intensity above threshold variance values, counted throughout the merged surface-most focal planes at each time point, and divided into weak, medium and strong sCMFs based on relative pixel intensity (S6 Fig and S5 Movie). We suspect that these sCMFs represent places in which multiple microtubules in some way overlap or cross over each other within the sub-cortical region and thus provide an assessment of microtubule organization. We observed significant decreases in the number of medium sCMFs, and in the total number of all sCMFs, after KNL-1 and BUB-1 RNAi knockdowns and in *cls-2 (or1948)* oocytes, compared to control oocytes (Fig 7).

While the KNL-1/BUB-1/CLS-2 sub-cortical patches are undetectable by the beginning of anaphase B (Fig 2A), we detected fewer sCMFs throughout anaphase B in mutant oocytes (Fig 7). Thus, if the KNL-1/BUB-1/CLS-2 patches do influence sub-cortical microtubule organization, whether they do so prior to anaphase B, or during anaphase B with undetected protein levels, requires further investigation. We conclude that KNL-1, BUB-1 and CLS-2 all promote a sub-cortical microtubule organization that includes more variability in the local microtubule distribution or arrangement during meiosis I anaphase B.

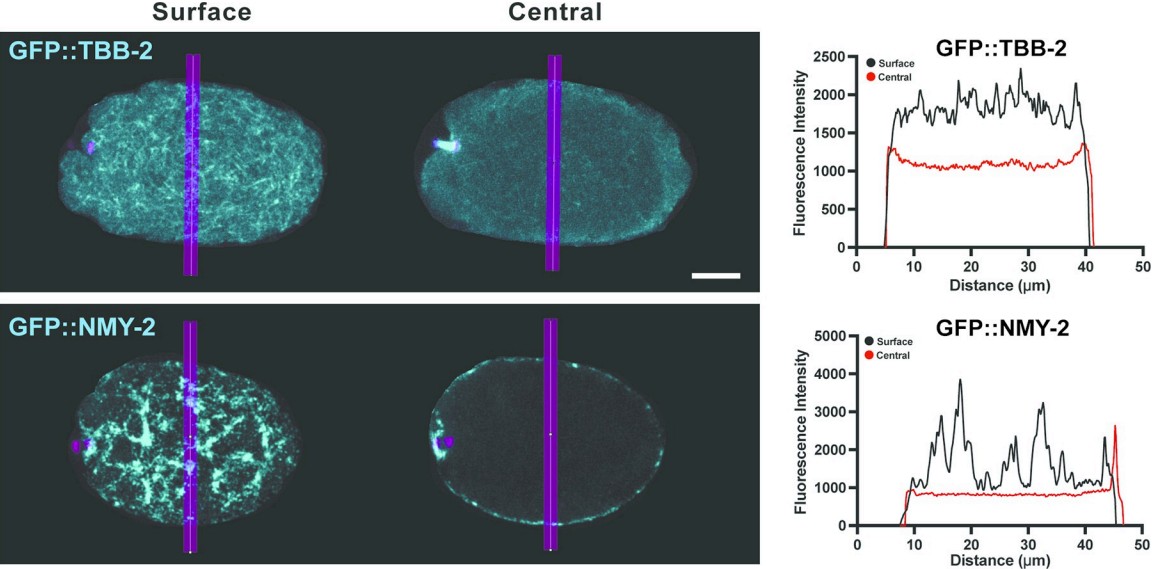

**Fig 5. Microtubule and NMY-2 distribution during meiosis I polar body extrusion.** Merged maximum intensity projections (MIPs) during meiosis I anaphase B of *ex utero* oocytes expressing GFP::TBB-2 (cyan) to mark microtubules (top row) or GFP::NMY-2 (cyan) to mark cortical actomyosin (bottom row), and mCherry::H2B (magenta) to mark chromosomes. MIPs of five surface-most focal planes (left column) or five central focal planes (middle column) are merged with MIPs of five consecutive internal focal planes that encompass most of the oocyte chromosomes, visible at the left, anterior oocyte end. Surface (black) and central (red) line scans of measured fluorescence intensity (right column) are from vertical magenta lines in oocyte images. See S3 File for raw data on line scans. Scale bar = 10 μm.

We next asked if the KNL-1/BUB-1/CLS-2 sub-cortical patches depend on microtubules for their localization or dynamics. The patches and the sub-cortical microtubules are both enriched near but underlying the oocyte cortex (Figs 3B and 5), and linear elements are known to include multiple proteins with microtubule binding activities [35]. However, KNL-1 linear elements were shown not to co-localize with microtubules, although that observation was made in a fixed oocyte at a single time point [15]. To determine if the sub-cortical patches depend on microtubules for their proper localization or dynamics, we used nocodazole to deplete oocyte microtubules (see Materials and Methods). First, we imaged *ex utero* oocytes that express GFP::TBB-2 and mCherry::H2B to confirm that our nocodazole treatments depleted spindle and cortical microtubules. In control oocytes, all appeared normal but upon exposure to nocodazole, microtubule levels were reduced, both in association with oocyte chromosomes and throughout the cortex (Figs 8A, S7C and S7D and S4 Movie). Next, we imaged *ex utero* oocytes that express GFP::KNL-1 to mark the sub-cortical patches and mCherry::H2B to mark chromosomes. We found to our surprise that upon exposure of oocytes early in meiosis I to nocodazole, the KNL-1 linear elements appeared longer and more numerous than those in control oocytes, although as in control oocytes they dissipated by the beginning of anaphase B (S6 Movie). We conclude that KNL-1, BUB-1 and CLS-2 not only influence the organization of sub-cortical microtubules but also depend on microtubules for their normal size and distribution. How the sub-cortical patches are anchored in proximity to the cell cortex in the absence of microtubules, and their potential relationship to cortical acto-myosin, will require further investigation (see Discussion).

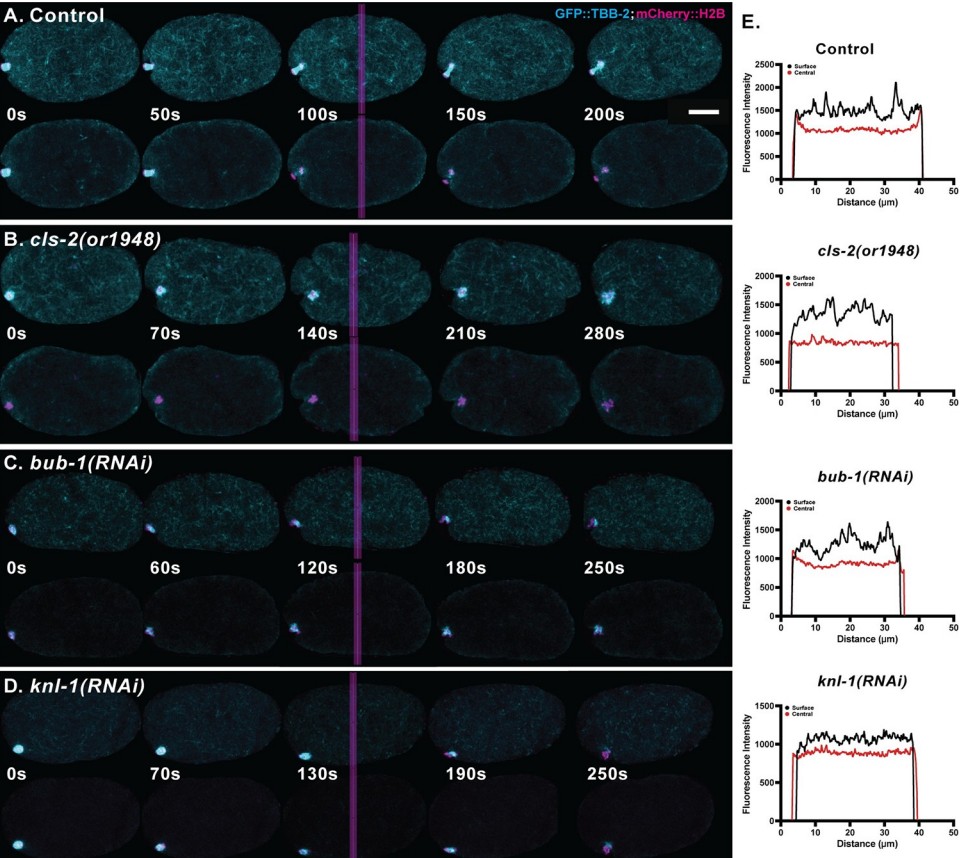

**Fig 6. Sub-cortical microtubules appear more evenly distributed in *knl-1*, *bub-1* and *cls-2* mutant oocytes.** Merged maximum intensity projections (MIPs) during meiosis I anaphase B of *ex utero* oocytes expressing GFP::TBB-2 (cyan) and mCherry::H2B (magenta) to mark microtubules and chromosomes, respectively. MIPs of five surface-most focal planes (upper rows) or five central focal planes (lower rows) showing microtubules are merged with MIPs of five consecutive internal focal planes that encompass most of the oocyte chromosomes, visible at the left, anterior end of each oocyte. In control oocytes (A) many small foci within an extensive network of microtubules are present throughout the surface-most focal planes, and microtubules are enriched peripherally in the central focal planes. We observed an apparent reduction in brightness of the small foci, and a loss of peripheral enrichment, in all three mutants (B-D). (E) Surface (black) and central (red) lines scans from vertical magenta lines in oocyte images. Scale bar = 10 μm.

## Chemical alteration of microtubule levels influences oocyte membrane ingression and interferes with polar body extrusion

If a more uniform distribution of oocyte sub-cortical microtubules in *knl-1*, *bub-1* and *cls-2* mutant oocytes reduces sub-cortical stiffness and results in excessive membrane ingression throughout the oocyte, we reasoned that disrupting microtubules with nocodazole might also reduce stiffness and result in excessively deep membrane ingression. To assess the impact of nocodazole-mediated microtubule depletion on oocyte membrane ingression, we imaged *ex utero* oocytes that express GFP::PH and mCherry::H2B fusions to mark the membrane and chromosomes. In control oocytes, spindle-associated furrows ingressed during chromosome separation, with shallow furrows also appearing throughout the oocyte during anaphase B. After exposure to nocodazole, ingressions still appeared near the oocyte chromosomes, but the chromosomes failed to separate, polar body extrusion always failed, and abnormally deep membrane ingressions appeared throughout the oocytes (Figs 8A, 8B and S4B and S3 Movie). The mean ingression length and mean sum of all lengths per oocyte were significantly

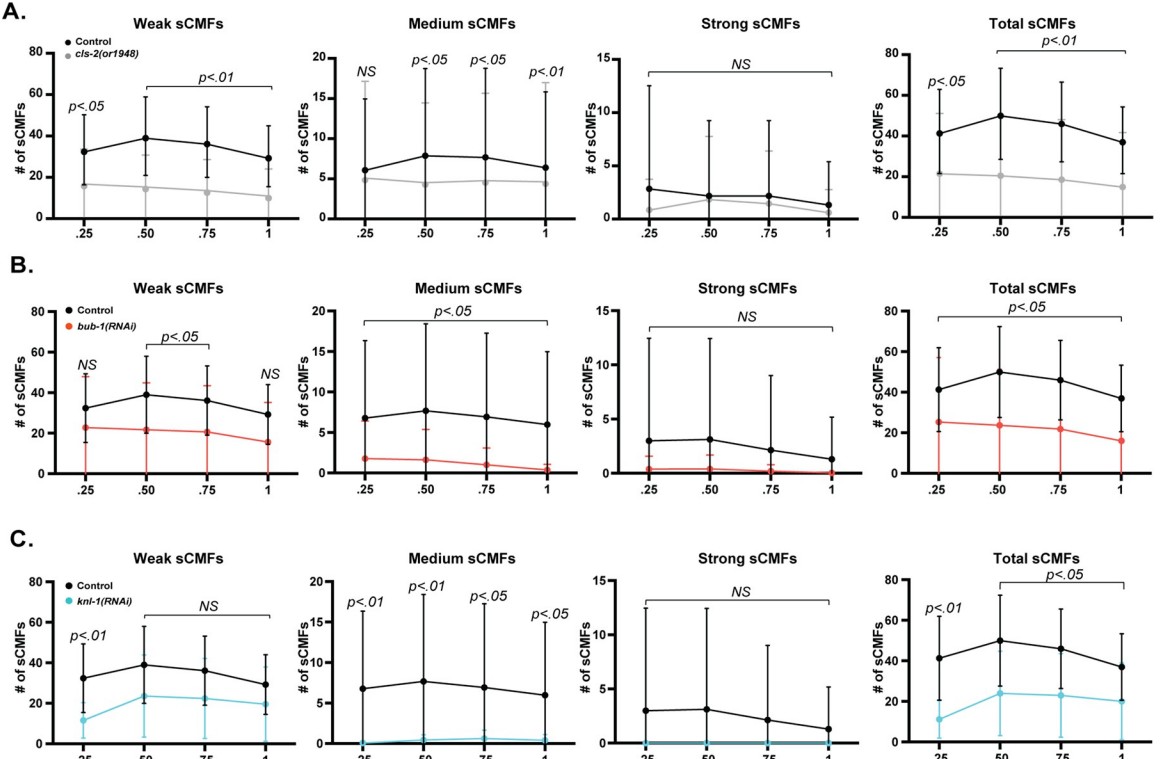

**Fig 7. The numbers of sub-cortical microtubule foci are reduced in *knl-1*, *bub-1* and *cls-2* mutant oocytes.** Comparisons of mean number of weak, medium and strong sub-cortical microtubule foci (sCMFs), and total sCMFs (sum of weak, medium, and strong sCMFs; see Materials and Methods), over normalized time during anaphase B in control versus *cls-2* (A), *bub-1* (B) and *knl-1* (C) mutant oocytes. The mean numbers of medium sCMFs, and total sCMFs, were significantly reduced in all three mutants. See S5 File for raw data used to calculate P values for sCMF differences. Scale bar = 10 μm.

increased relative to control oocytes, although ingression number was not affected (Fig 8C; see Discussion).

We next asked if chemically stabilizing oocyte microtubules with taxol would reduce the extent of membrane ingression during polar body extrusion. Taxol treatment variably elevated sub-cortical microtubule levels during meiosis I to an extent that was not statistically significant in our small sample size of five oocytes (Figs 8A, S7C and S7D and S3 Movie). Nevertheless, taxol treatment did significantly reduce the extent of membrane ingression during anaphase B (Fig 8A and 8B and S3 Movie), although polar bodies were still extruded in about 2/3 of taxol treated oocytes (Figs 8B and S4B). The increased and decreased membrane ingressions observed after nocodazole and taxol treatment, respectively, further support our hypothesis that sub-cortical microtubules limit membrane ingression during meiosis I polar body extrusion.

## Increased levels of sub-cortical microtubules suppress membrane ingression in *cls-2* mutants

Because an altered, more even distribution of sub-cortical microtubules in *cls-2* mutant oocytes correlates with excess membrane ingression, we next asked if increasing the levels of sub-cortical microtubules in *cls-2* mutants can suppress excess membrane ingression. Two microtubule regulators in *C. elegans*, the kinesin 13/microtubule depolymerase family member

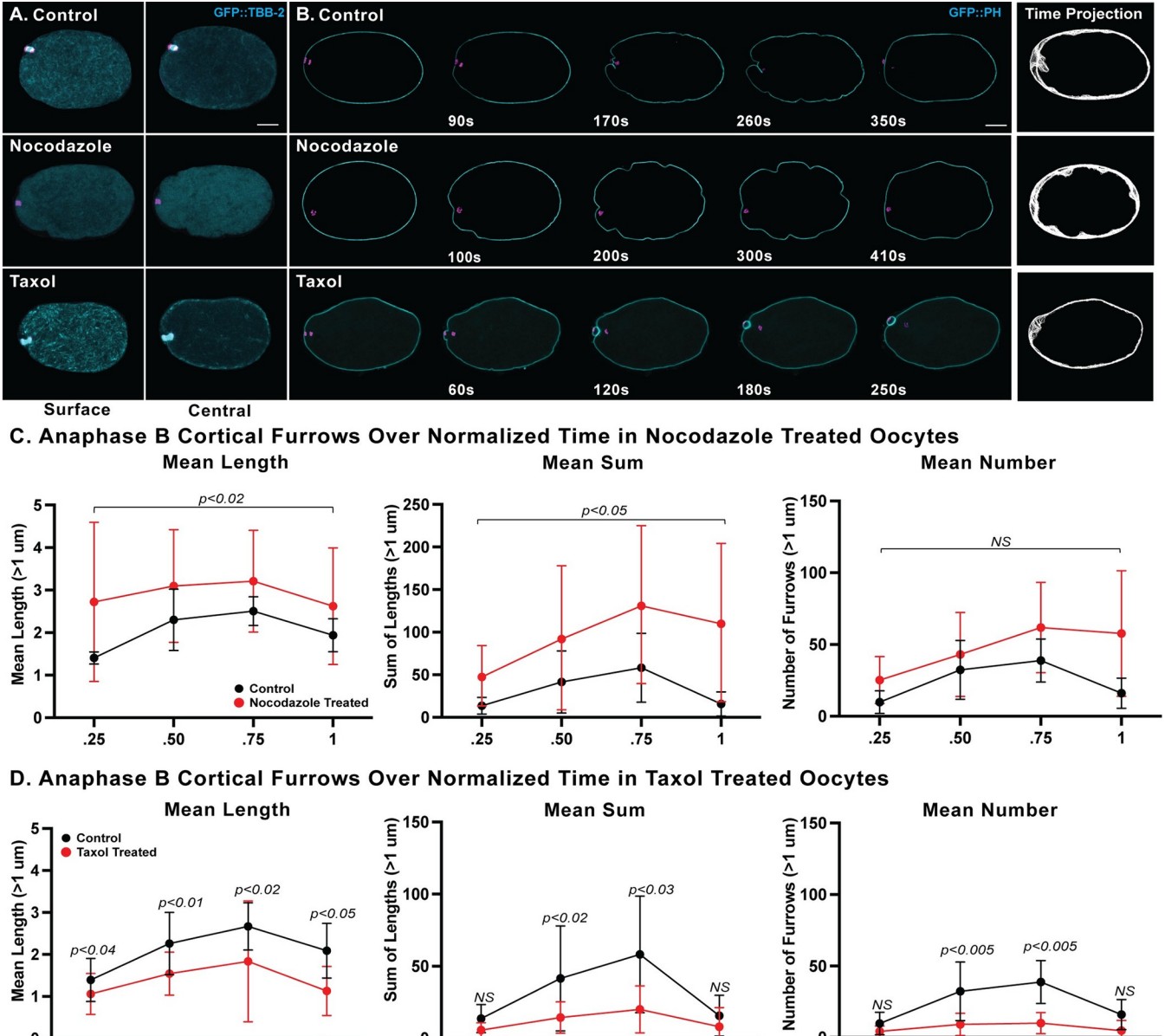

**Fig 8. Nocodazole and taxol modulate oocyte membrane ingression.** (A) Merged maximum intensity projections (MIPs) during meiosis I anaphase B of *ex utero* oocytes expressing GFP::TBB-2 (cyan) and mCherry::H2B (magenta) to mark microtubules and chromosomes, respectively, in control (top), 20 μg/ml nocodazole-treated (middle), and 200 nM taxol-treated (bottom) oocytes (see Materials and Methods). MIPs of five surface-most focal planes (left column) or five central focal planes (right column) showing microtubules are merged with MIPs of five consecutive internal focal planes that encompass most of the oocyte chromosomes, visible at the left, anterior ends of the oocytes. (B) Selected and merged focal planes of *ex utero* oocytes expressing GFP::PH (cyan) and mCherry::H2B (magenta) to mark membranes and chromosomes, respectively, in control (top row), nocodazole-treated (middle row), and taxol-treated (bottom row) oocytes. A single central focal plane is shown for the membrane, merged with MIPs of five consecutive internal focal planes that encompass most of the oocyte chromosomes. Time projections of single central focal planes for the membrane, depicting all anaphase B time points, are at far right. (C-D) Quantification of the mean length of membrane ingressions that were 1μm or more in length, the mean sum of lengths, and the mean number of membrane ingressions, over normalized anaphase B time in control and chemically treated oocytes. While polar body extrusion always failed after nocodazole treatment, polar body extrusion was usually successful after taxol treatment (arrowhead; also see S4B Fig). The mean length and mean sum of lengths, but not the mean number of ingressions were increased in nocodazole-treated oocytes, while both the mean length and number of ingressions were decreased in taxol-treated oocytes. See S7D Fig for quantification of microtubule levels after nocodazole and taxol treatments. Scale bar = 10 μm.

KLP-7, and the XMAP215/chTOG family member ZYG-9, both have been shown to limit spindle and cortically associated microtubules levels [24,25,36]. We first asked whether the increased levels of cortically associated microtubules in *klp-7* and *zyg-9* mutants depend on CLS-2, by quantifying sub-cortical microtubule foci (sCMFs) during anaphase B in *ex utero klp-7(RNAi)* and *zyg-9(RNAi)* single mutant, and *cls-2(or1948) klp-7(RNAi)* and *zyg-9(RNAi); cls-2(or1948)* double mutant oocytes, all expressing GFP::TBB-2 and mCherry::H2B. We observed significant increases in the microtubule signal (Fig 9A–9F and S4 Movie), and in total sCMF number (Fig 9H), relative to control oocytes, not only in *klp-7* and *zyg-9* single mutants, but also in the corresponding *cls*-2 double mutants.

While KLP-7 and ZYG-9 knockdowns both led to increased levels of sub-cortical microtubules, the microtubule properties in each mutant were distinct. While the number of sCMFs in *zyg-9* mutants did not show a statistically significant dependence on CLS-2, the sCMFs were partially but significantly reduced in *klp-7 cls*-2 double mutants relative to *klp-7* single mutants (Fig 9H). Nevertheless, sCMFs in both double mutants were still significantly elevated compared to control and *cls-2* mutants (Fig 9H). Finally, the distribution of cortical microtubules appeared to differ qualitatively in *klp-7* and *zyg-9* single mutants, and in the corresponding double mutants. The cortical microtubules in *klp-7* mutant oocytes appeared more evenly distributed (Fig 9C and S4 Movie), whereas the cortical microtubules in *zyg-9* mutant oocytes formed many small puncta (Fig 9E and S4 Movie).

After determining that KLP-7 or ZYG-9 depletion elevates sub-cortical microtubule levels in *cls-2* mutant oocytes, we next asked whether the excess membrane ingression in *cls-2* mutants is suppressed in *cls-2 klp-7* and *zyg-9; cls-2* double mutants, imaging oocytes that express GFP::PH and mCherry::H2B (Fig 10A–10F and S3 Movie). We observed significant reductions in both the mean ingression length and the mean sum of lengths in *cls-2 klp-7* and *zyg-9; cls-2* double mutants compared to *cls-2* single mutant oocytes (Fig 10G and 10H). As a final test of whether increased microtubule levels can suppress the excess membrane ingression, we used taxol treatment to stabilize microtubules in *cls-2(or1948)* mutant oocytes and also observed a significant reduction in the extent of membrane ingression (S7 Fig). These results support our hypothesis that sub-cortical microtubules act to limit oocyte membrane ingression during meiosis I polar body extrusion.

## Altered microtubule dynamics interfere with polar body extrusion

To further explore the relationship between cytoskeletal regulation throughout the oocyte and polar body extrusion, we also scored whether polar body extrusion succeeded or failed in *cls-2*, *klp-7* and *zyg-9* single and double mutant oocytes. While the elevated microtubules levels in *cls-2 klp-7* and *zyg-9; cls-2* double mutants correlated with suppression of excess membrane ingression, polar body extrusion was not rescued. Instead, it was equally or more defective in the double mutants compared to *cls-2*, *klp-7* and *zyg-9* single mutants. Indeed, polar body extrusion only rarely succeeded even in all single mutant oocytes (S4B Fig), which all have extensive oocyte spindle assembly defects [24–26]. Furthermore, although membrane ingression in both *klp-7* and *zyg-9* mutant oocytes generally appeared reduced, we nevertheless observed deep and prominent ingressions near the oocyte chromosomes in some *cls-2*, *klp-7* and *zyg-9* single mutants (Figs 10C, 10E and S3 and S3 Movie). These prominent furrows were absent in *cls-2 klp-7* and in *zyg-9; cls*-2 double mutants (Figs 10D, 10F and S3 and S3 Movie). Finally, polar body extrusion always failed after nocodazole treatment to destabilize microtubules but failed in only about 1/3 of oocytes exposed to taxol to stabilize them (S4B Fig). However, our taxol treatments may not have elevated microtubules sufficiently to interfere with polar body extrusion in some oocytes (S7D Fig). Though variable, these results are consistent

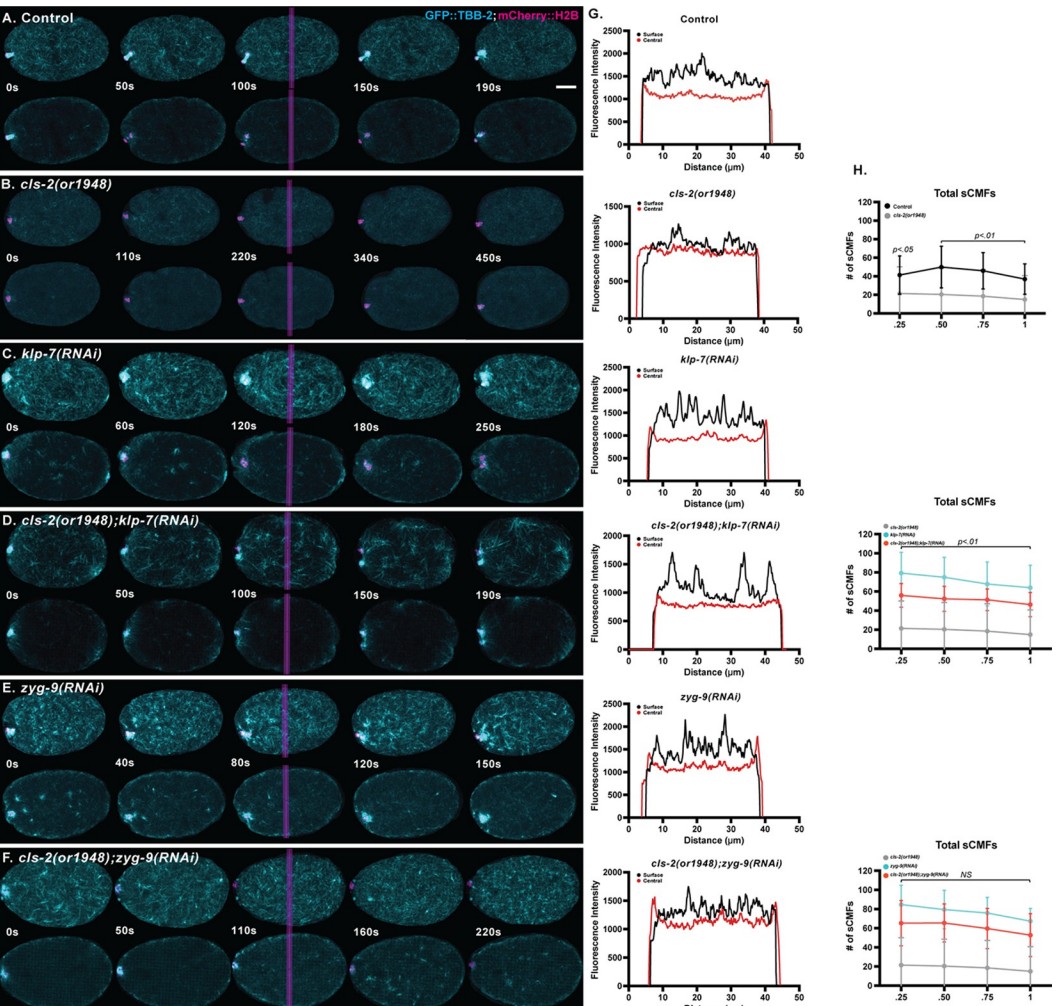

**Fig 9. Loss of KLP-7/kinesin-13 or ZYG-9/chTOG increases sub-cortical microtubules in *cls-2(or1948)* oocytes.** (A-F) Merged maximum intensity projections (MIPs) during meiosis I anaphase B in live *ex utero* control and mutant oocytes expressing GFP::TBB-2 (cyan) and mCherry::H2B (magenta) to mark microtubules and chromosomes, respectively. MIPs of five surface focal planes (upper rows) or five central focal planes (lower rows) showing microtubules are merged with MIPs of five consecutive central focal planes that encompass most of the oocyte chromosomes. Spindle-associated microtubules and chromosomes are visible at the left, anterior end of each oocyte. (G) Surface (black) and central (red) lines scans are from vertical magenta lines in oocyte images. (H) Total sub-cortical microtubule foci (sCMF) numbers over normalized anaphase B time. Total sCMFs were significantly reduced in *cls-2(or1948)* mutants, and significantly increased in *klp-7(RNAi)* and *zyg-9(RNAi)* single and double mutants, compared to control oocytes. Total sCMFs were partially but significantly decreased in *cls-2(or1948) klp-7* (RNAi) double mutants compared to *klp-7(RNAi)* single mutants, but not in *zyg-9(RNAi); cls-2(or1948)* double mutants compared to *zyg-9(RNAi)* single mutant oocytes. Scale bar = 10 μm.

with sub-cortical microtubule organization and levels being important for polar body extrusion in *C. elegans* oocytes.

## Microtubules influence the distribution of cortical actomyosin in *C. elegans* oocytes

We previously hypothesized that cortically associated microtubules themselves, through the biophysical properties of their tubular polymeric structure, directly confer a stiffness that limits the extent of membrane ingression during polar body extrusion [11]. Alternatively, the

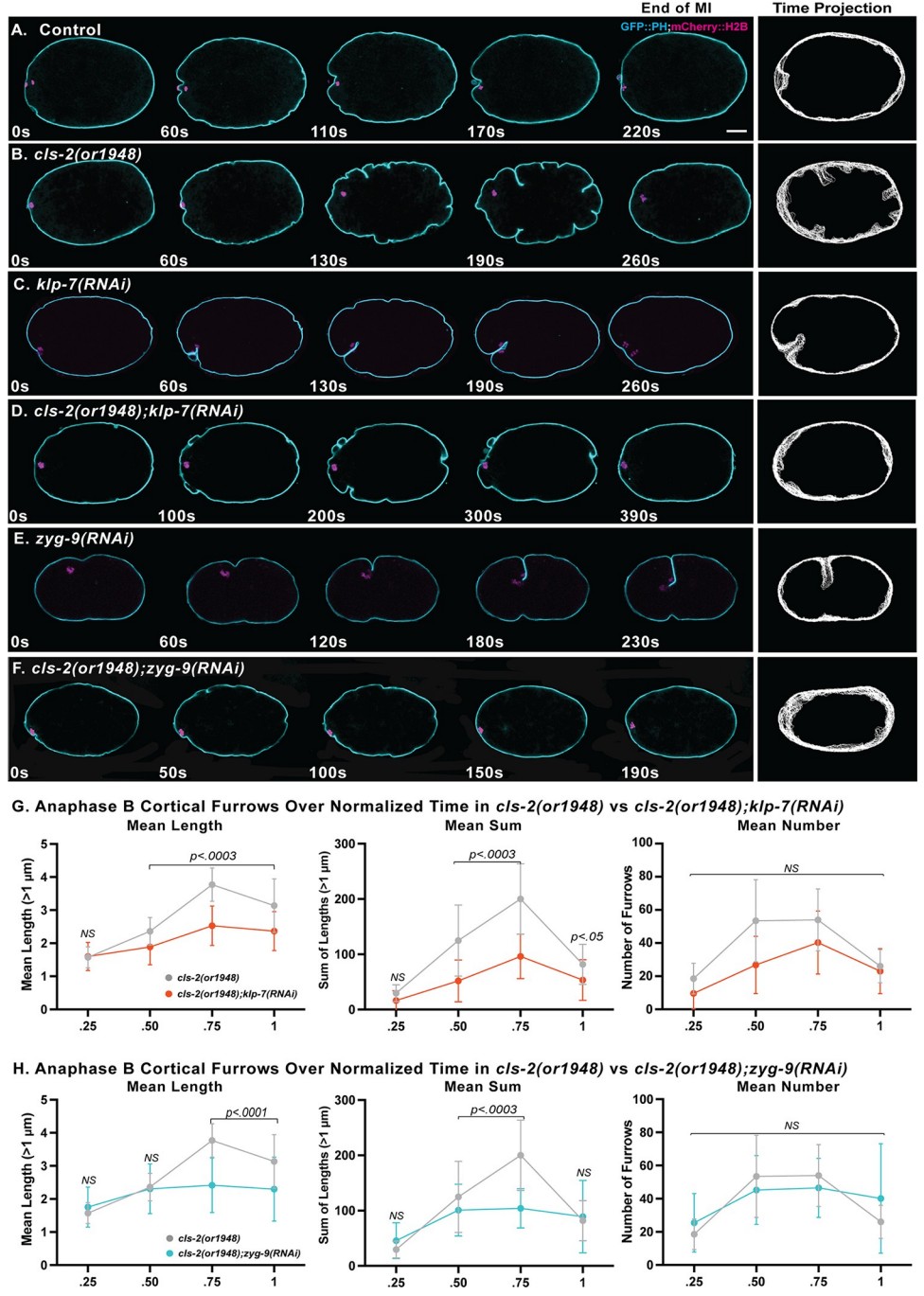

**Fig 10. KLP-7/kinesin-13 or ZYG-9/chTOG knockdown suppresses excess membrane ingression in *cls-2(or1948)* oocytes.** (A-F) Selected and merged focal planes during meiosis I anaphase B in live *ex utero* control and mutant oocytes expressing GFP::PH (cyan) and mCherry::H2B (magenta) to mark membranes and chromosomes, respectively. A single central focal plane is shown for the membrane, merged with a maximum intensity projection of five consecutive focal planes that encompass most of the oocyte chromosomes, visible at the left, anterior end of each oocyte. Anaphase B time projections of single central focal planes for the membrane are at far right. (G&H) Quantification of the mean length of membrane ingressions, the mean sum of lengths, and the mean number of membrane ingressions, over normalized time in control and mutant oocytes. The mean length and the mean sum of lengths, but not the mean number of ingressions, are significantly reduced in the double mutants compared to *cls-2 (or1948)* single mutant oocytes. Scale bar = 10 μm.

cortically associated microtubules might indirectly influence membrane ingression by signaling to cortical actomyosin in ways that we failed to detect.

To explore whether microtubules themselves, or microtubule signaling to cortical actomyosin, limits membrane ingression, we imaged the dynamics of a GFP fusion to the non-muscle myosin II NMY-2 (GFP::NMY-2) in control and mutant *ex utero* oocytes. To our surprise, we observed reproducible differences in the distribution of cortical GFP::NMY-2, just before and during polar body extrusion in both *klp-7* and *zyg-9* mutants, compared to control oocytes. In control oocytes, we observed the expected network of evenly distributed GFP::NMY-2 foci that dissipated during anaphase B (Figs 11A and S8 and S7 Movie). In nearly all *klp-7* and *zyg-9* mutant oocytes, and in *cls-2 klp-7* and *zyg-9; cls-2* double mutants, we observed bright linear arrays of GFP::NMY-2 foci, often but not always oriented in parallel to each other and transverse to the long axis of the oocyte (Figs 11C–11E, S10 and S11 and S7 Movie). While we have not quantified these differences, the defects appeared most severe just before and at the beginning of anaphase B, and they may be more severe in *zyg-9* mutant oocytes (Figs 11 and S8–S11). Moreover, we observed less frequent and less prominent but nevertheless similar defects

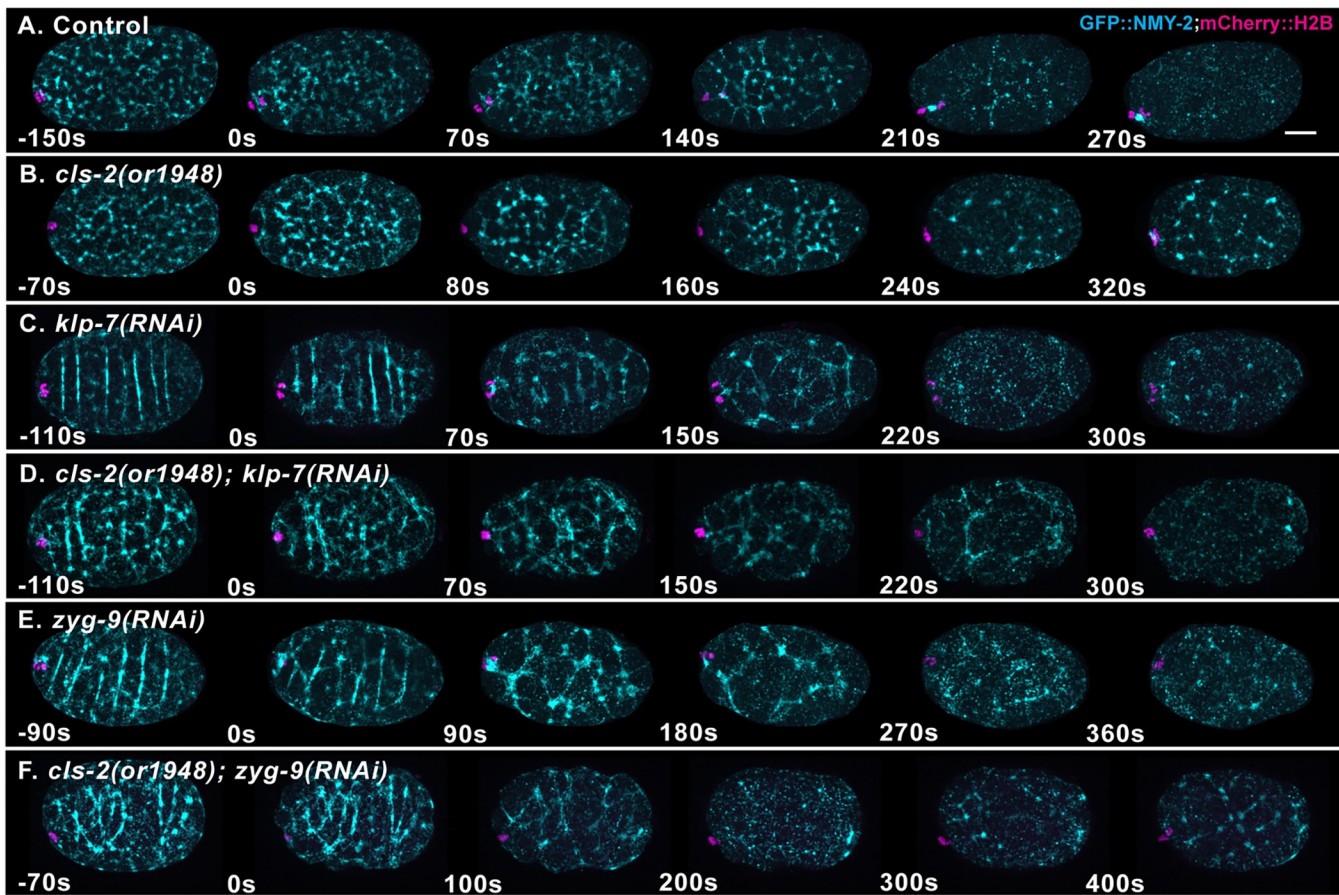

**Fig 11. Altered sub-cortical microtubules are associated with abnormal distributions of cortical non-muscle myosin II.** (A-F) Merged maximum intensity projections (MIPs) during meiosis I anaphase B of live *ex utero* control and mutant oocytes expressing NMY-2::GFP (cyan) and mCherry::H2B (magenta) to mark non-muscle myosin II and chromosomes, respectively. MIPs of five surface-most focal planes showing non-muscle myosin II are merged with MIPs of five consecutive internal focal planes that encompass most of the oocyte chromosomes, visible at the left, anterior end of each oocyte. The GFP::NMY-2 foci formed extended and sometimes transversely oriented and parallel linear arrays in *klp-7* and *zyg-9* mutant oocytes, most prominently just before and at the beginning of anaphase B. The NMY-2::GFP distribution appeared more normal in *cls-2or1948)* oocytes, but similarly abnormal and extended NMY-2 arrays also were observed in some *cls-2* mutant oocytes (S11 Fig). Scale bar = 10 μm.

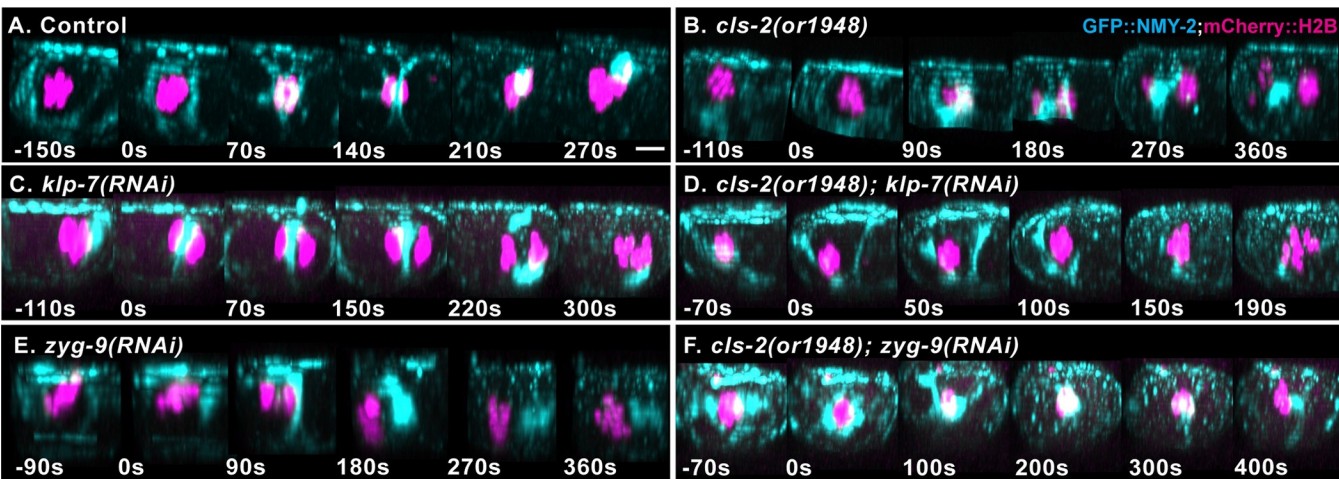

**Fig 12. Contractile ring dynamics during meiosis I polar body extrusion are defective in *cls-2*, *klp-7* and *zyg-9* mutant oocytes.** Merged maximum intensity projections of all sixteen focal planes after Imaris-mediated rotation to obtain ring-centric, more oocyte-end-on views of contractile ring dynamics in control and mutant *ex utero* oocytes expressing NMY-2::GFP (cyan) and mCherry:H2B (magenta) to mark non-muscle myosin II and chromosomes, respectively. A ring forms early in anaphase B and constricts in control oocytes, but rings either are not apparent or are unstable in mutant oocytes, with NMY-2::GFP near the chromosomes often merging into one or a few bright puncta. Scale bar = 5 μm.

in some *cls-2* mutant oocytes (S9 Fig). While the spatial organization of NMY-2 foci was altered in mutant oocytes, their dissipation during anaphase B appeared similar to controls (Figs 11 and S8–S11). These observations raise the possibility that sub-cortical microtubules might act indirectly through actomyosin to limit membrane ingression (see Discussion).

After documenting defects in both cortical NMY-2 dynamics and polar body extrusion in *klp-7*, *zyg-9* and *cls-2* mutant oocytes, we next examined contractile ring assembly and ingression in control and mutant oocytes. In our previous study, we found that *cls-2* mutants have early defects in ring assembly or stability [11]. To inspect the penetrant polar body extrusion defects in *cls-2*, *klp-7* and *zyg-9* mutant oocytes (S4B Fig), we used Imaris software to rotate and project three-dimensional renderings of our NMY-2::GFP live imaging data and thereby observe contractile ring dynamics from a more oocyte end-on and ring-centric view (Figs 12 and S12 and S8 Movie).

We observed similar early and penetrant defects in meiosis I contractile ring assembly or stability in *cls-2*, *klp-7* and *zyg-9* single mutants, and in both double mutants. GFP::NMY-2 foci often did not form detectable rings, and even when rings were detected they were unstable, with GFP::NMY-2 moving into one or a few bright foci near the chromosomes in most oocytes and ultimately failing to extrude polar bodies (Figs 12, S4B and S12 and S8 Movie). We have not quantified these variable and dynamic contractile ring defects in detail, but depleting any one of three different regulators of oocyte microtubules results in early and penetrant contractile ring assembly or stability defects, highlighting the importance of microtubule regulation during *C. elegans* polar body extrusion.

## Discussion

We have shown that two subunits of an outer kinetochore sub-complex—the kinase BUB-1 and the CLASP family member CLS-2—along with the outer kinetochore scaffolding protein KNL-1, co-localize both to linear elements associated with the oocyte meiosis I spindle, and to related sub-cortical patches that are mobile and distributed throughout the oocyte, underlying its cortex until they dissipate at the beginning of anaphase B during *C. elegans* meiosis I cell division. As previously documented at kinetochores, CLS-2 localization to these sub-cortical

patches required BUB-1 and KNL-1. However, unlike at kinetochores, we also observed extensive mutual dependence for sub-cortical patch localization. This analysis relied on partial depletions of these proteins, as KNL-1 and BUB-1 are both required for fertility and cannot be analyzed in null oocyte backgrounds. Thus, our results may not fully reveal either the dependencies of these proteins on each other either for patch localization, or their other requirements. Nevertheless, knocking down any one of these sub-cortical patch components resulted both in an altered distribution of sub-cortical microtubules and in excess membrane ingression throughout the oocyte during polar body extrusion. Moreover, nocodazole and taxol treatments, to destabilize and stabilize microtubules, respectively, led to increased and decreased membrane ingression. Finally, genetic backgrounds with increased levels of sub-cortical microtubules, and taxol treatment to stabilize microtubules, both suppressed the excess membrane ingression in *cls-2* mutant oocytes. Although we cannot rule out microtubules indirectly opposing membrane ingression by signaling to actomyosin (see below), these results support our hypothesis that sub-cortical microtubules confer a stiffness that limits the extent of cortical actomyosin-mediated membrane ingression throughout the oocyte during meiosis I polar body extrusion.

## CLS-2/CLASP mediates only a subset of the KNL-1/BUB-1 regulation of oocyte membrane ingression

Because CLS-2/CLASP family members are known to promote microtubule stability [12–14], we anticipated that CLS-2 might mediate all sub-cortical microtubule regulation by the KNL-1/BUB-1/CLS-2 sub-cortical patches. However, we found that while RNAi knockdown of either KNL-1 or BUB-1 increased both the number of ingressions and also the mean length and the mean sum of all ingression lengths, eliminating CLS-2 increased only the mean length and the mean sum of lengths, but not the number of ingressions. This difference is significant because we observed the more extensive requirements for KNL-1 and BUB-1 after RNAi knockdowns that only partially eliminated gene function, while for CLS-2 we quantified membrane ingression in oocytes from worm homozygous for the null allele *cls-2(or1948)*. We conclude that CLS-2 mediates only a subset of the KNL-1/BUB-1/CLS-2 functions that limit oocyte membrane ingression during polar body extrusion. Putative KNL-1/BUB-1-dependent but CLS-2 independent factors that limit the number of membrane ingressions might, like CLS-2, influence sub-cortical microtubules, but in a distinct manner that limits the number rather than the extent of ingressions.

Arguing against a role for sub-cortical microtubules affecting the number of ingressions, we observed an increase only in the length but not the number of ingressions after nocodazole treatment to destabilize oocyte microtubules. However, after taxol treatment to stabilize microtubules, we observed a significant decrease in both the length and number of ingressions. Thus, it is possible that, compared to the indiscriminate reduction caused by nocodazole treatment, altered sub-cortical microtubule dynamics in other mutant backgrounds could influence ingression number.

Alternatively, KNL-1 and BUB-1 might act through cortical actomyosin to limit the number of ingressions independently of CLS-2. One potential target for such regulation is the kinase CSNK-1, a negative regulator of the non-muscle myosin II NMY-2, which when reduced in function, was reported to result in large and more persistent cortical NMY-2 foci and excess membrane ingression throughout the oocyte during polar body extrusion [21]. Further investigation of CSNK-1, and other NMY-2 regulators, might provide further insight.

Several other *C. elegans* kinetochore components also are present in linear elements. These include HCP-1/2, HIM-10, KNL-3, KBP-1, NDC-80, ZWL-1, SEP-1 and ROD-1

[15,16,30,33,37]. Presumably, these all co-localize with the KNL-1/BUB-1/CLS-2 sub-cortical patches, but whether they all influence membrane ingression or other processes during oocyte meiosis is not known. Finally, in *Drosophila*, the Ald ortholog of the conserved kinetochore-associated kinase Mps1 is present in filamentous structures during oocyte meiotic cell division, and these filaments also contain other kinetochore proteins including BUBR1 and *polo* [38]. While Mps1 is not conserved in *C. elegans*, it is intriguing that similar structures with kinetochore components have been reported in *Drosophila* oocytes.

### Do sub-cortical microtubules directly or indirectly limit oocyte membrane ingression during polar body extrusion?

Our hypothesis that stiffness provided by sub-cortically enriched microtubules directly limits membrane ingression during polar body extrusion in *C. elegans* is supported by the following observations: (i) oocyte microtubules are enriched within a sub-cortical layer throughout the oocyte, which places them in a position to resist cortical actomyosin-driven membrane ingression; (ii) nocodazole treatment to destabilize microtubules, and the more evenly distributed sub-cortical microtubules in *cls-2* mutant oocytes, increased membrane ingression throughout the oocyte; (iii) taxol treatment to stabilize microtubules decreased membrane ingression; and (iv) increased levels of sub-cortical microtubules in *cls-2 klp-7* and *zyg-9; cls-2* double mutants, and in taxol treated *cls-2* mutant oocytes, suppressed the excess membrane ingression in *cls-2* mutants. The tubular structure of microtubules makes them relatively stiff compared to microfilaments [39], and it seems plausible that, in sufficient numbers, with appropriate length distributions and organization, and perhaps with cross-linking between themselves or with microfilaments (see below), sub-cortical microtubules could confer a stiffness that resists cortical actomyosin-driven membrane ingression.

While the sub-cortical microtubules might themselves resist membrane ingression, crosstalk between the microtubule and microfilament cytoskeleton is extensive [9,10], and our results do not exclude the more conventional possibility that sub-cortical microtubules indirectly suppress membrane ingression by acting through cortical actomyosin. In support of microtubules signaling to actomyosin, we observed changes in the organization of the cortical non-muscle myosin II NMY-2 in both *klp-7*/kinesin-13 and *zyg-9*/chTOG mutants that might account for the decreased membrane ingression observed in *cls-2 klp-7* and *zyg-9; cls*-2 double mutants relative to *cls-2* single mutants.

The abnormal distributions of cortical NMY-2 foci in *klp-7* and *zyg-9* mutant oocytes clearly indicate that alterations in sub-cortical microtubule dynamics influence cortical actomyosin dynamics. However, to our knowledge, it is not known how the distribution and location of cortical NMY-2 foci outside the contractile ring correlate with sites of membrane ingression, and it is not obvious how the arrangement of cortical NMY-2 into prominent linear arrays in *klp-7* and *zyg-9* mutants, relative to the more dispersed network of NMY-2 foci in control oocytes, might suppress membrane ingression, rather than promote more extensive ingression in the vicinity of the linear NMY-2 arrays. Moreover, the abnormal NMY-2 distributions were most apparent before and at the beginning of anaphase B, while oocyte membrane ingression peaked late in anaphase B, and we observed similar though less prominent alterations in NMY-2 distribution in *cls-2* mutant oocytes, which have increased membrane ingression. We therefore favor an alternative view that, rather than suppressing membrane ingression, the abnormal arrays of NMY-2 foci in these mutant oocytes are an indirect consequence of altered, microtubule-mediated sub-cortical stiffness. Further investigation with live imaging to examine the spatial and temporal relationships of cortical NMY-2 foci, KNL-1/BUB-1/CLS-2 sub-cortical patches, sites of membrane ingression, and microtubule

distribution throughout meiosis I cell division will improve our understanding of these regulatory relationships.

## Anchoring of cortical actomyosin and of sub-cortical microtubules

A key property of cortical actomyosin is its attachment to the overlying plasma membrane, which links tension and elasticity in the cell cortex to the membrane, and to cell shape and morphogenesis [1,4]. One important mechanism for attachment to the membrane occurs through Ezrin/Radixin/Moesin (ERM) family members that bind both to microfilaments and to a variety of membrane proteins [40,41]. The increased cross-linking of cortical microfilaments to the membrane by ERM family members, during insect and animal cell division, stiffens the cortex to promote mitotic rounding [6–8]. Myosin 1 and filamin family members also have been implicated in attaching cortical microfilaments to the overlying membrane in some cell types [41–44].

The conservation of cortex/membrane attachment by ERM, Myosin 1, and filamin family members in different animal phyla and cell types has not been extensively explored. While we have shown that cortical actomyosin is coupled to the membrane in early embryonic blastomeres in *C. elegans* [45], the molecules that mediate attachment are not known. The *C. elegans* genome encodes only a single ERM family member, MOE-1, with limited expression in several larval cell types and limited post-embryonic requirements during larval intestinal lumen and vulval development [46]. Similarly, the two Myosin 1 family members in *C. elegans*, HUM-1 and -5, and the two *C. elegans* filamin family members, FLN-1 and -2, are expressed outside of the germline in a limited number of cell types and have limited post-embryonic requirements [47–49]. Further investigation is needed to identify *C. elegans* factors that attach cortical microfilaments to the cell membrane, not only in oocytes but throughout development.

How oocyte microtubules and the KNL-1/BUB-1/CLS-2 patches are restricted to a sub-cortical layer throughout the oocyte, and whether these microtubules and sub-cortical patches extend into and intermingle with or are molecularly linked to cortical actomyosin or to the cell membrane, also is not clear. One appealing candidate for linking sub-cortical microtubules to cortical actomyosin is dynein, which has been implicated in coordinating interactions between the microtubule and microfilament cytoskeletons in other contexts, including the early *C. elegans* embryo [50,51]. Whether *C. elegans* dynein influences the distribution of oocyte sub-cortical microtubules or the KNL-1/BUB-1/CLS-2 sub-cortical patches, or affects membrane ingression during polar body extrusion, requires further investigation.

## Additional cortical roles for microtubules in *C. elegans* and other eukaryotes

Microtubules may have additional cortical roles later in *C. elegans* embryogenesis. During mitosis, astral microtubules in one-, two- and four-cell stage embryos stop growing when their plus ends reach the cell cortex, with plus-end capture by complexes that include dynein mediating mitotic spindle positioning during the first few embryonic cell divisions [35]. The termination of microtubule plus-end growth in early embryos requires the cortically localized GTPase EFA-6, which is no longer detectable after the 4-cell stage [52]. In *efa-6* null mutants, and in wild-type embryos beyond the 4-cell stage, astral microtubules continue to elongate upon reaching the cell cortex, forming extensive networks that occupy or are in close proximity to early embryonic cell cortices [52]. While it is not known whether the ubiquitous cortically associated microtubules observed in wild-type embryos after the 4-cell stage are important for normal development, cell contact can orient cell division axes in *C. elegans* embryos by modulating cortical actomyosin dynamics [45]. Perhaps microtubule-mediated

stiffness, together with contact-mediated, actomyosin-driven orientation of division axes, are in part responsible for generating the remarkably invariant pattern of cell division axes that constitute *C. elegans* embryogenesis [53].

Microtubules may also influence cortical stiffness in other eukaryotic cell types. In developing plant cells, aligned bundles of cortical microtubules are required for proper cell shape and morphogenesis [54,55], although these bundles may act indirectly by mediating the secretion and organization of extracellular cellulose fibrils that ultimately form the rigid plant cell walls. Notably, microtubules form a cortical band in human platelets that is required for their flattened, discoid shape [56,57], and in a study of T-cell migration, stabilization of microtubules by taxol, while at the same time inhibiting non-muscle myosin II contractility, led to increased cortical stiffness [58]. Furthermore, microtubules might contribute to cortical stiffness even in animal phyla that employ ERM family members during mitotic rounding. In an impressively high-throughput Atomic Force Microscopy screen of all human kinases for cortical stiffness factors [59], the only kinase identified, other than the SILK-1 kinase that phosphorylates ERM proteins to activate microfilament cross-linking to the membrane [7,8], was BUB-1. Thus, a BUB-1 ortholog influences cortical stiffness in a cell type that uses ERM proteins during mitotic rounding, perhaps through microtubule regulation. It will be interesting to test whether CLASP orthologs also influence cortical stiffness in other animal cell types.

## Microtubules and polar body extrusion in mammals and in *C. elegans*

While actomyosin appears to play very different roles during *C. elegans* oocyte meiotic spindle assembly and positioning compared to vertebrate oocytes (see S6 File), microtubules are important for polar body extrusion in both. Even though chromosomes, or even just DNA-coated beads, can induce the formation of an actin cap and a surrounding contractile ring in mouse oocytes, spindle microtubules nevertheless are required for the successful completion of polar body extrusion [60]. In *C. elegans*, the requirements we have documented for CLS-2/CLASP, KLP-7/kinesin-13, and ZYG-9/chTOG, and the consequences of nocodazole and taxol treatment, all indicate that microtubules play important role(s) in polar body extrusion. Further investigation of microtubule and microfilament dynamics in both vertebrate and invertebrate oocytes is needed to better understand the similarities and differences in the mechanisms that mediate oocyte meiotic cell division across animal phyla.

## The relationship between oocyte membrane ingression and polar body extrusion

While our results clearly indicate that oocyte microtubules influence membrane ingression throughout the oocyte during polar body extrusion, the functional importance of this regulation is much less clear. Because the contractile ring during *C. elegans* oocyte meiotic cytokinesis forms within a much larger and actively contractile cortical actomyosin network (Fig 1), we speculate that a proper balance of cortical actomyosin tension and sub-cortical microtubule stiffness modulates cortical actomyosin dynamics throughout the network to promote contractile ring assembly and ingression. Indeed, in *klp-7*, *zyg-9* and *cls-2* mutant oocytes, a contractile ring often begins to assemble but then becomes distorted and usually collapses into a single prominent furrow that ultimately regresses with failed polar body extrusion. It is tempting to infer that altered sub-cortical stiffness in mutant oocytes puts a strain on the cortical actomyosin network that disrupts contractile ring assembly and ingression.

If our hypothesis that strain within the cortical actomyosin network disrupts contractile ring dynamics were correct, we would expect mutant oocytes with similarly excessive membrane ingression to fail in polar body extrusion with similar penetrance. However, while we

observed quantitatively similar membrane ingression defects in *rod-1*, *knl-1*, *bub-1* and *cls-2* mutant oocytes, they differed substantially with respect to the penetrance of their polar body extrusion defects. Over 1/3 of null *cls-2(or1948)* oocytes succeeded in polar body extrusion as we scored it, while only 10% of *bub-1(AID)* oocytes succeeded, and 90% or *rod-1(RNAi)* oocytes and 75% of *knl-1(AID)* oocytes succeeded in polar body extrusion. More complete depletion of these proteins, and a detailed examination of oocyte meiotic contractile ring dynamics, might reveal shared defects. Nevertheless, the similar membrane ingression defects, in mutants with very different rates of failure in polar body extrusion, argues against excess strain throughout the cortical actomyosin network interfering with polar body extrusion.

Another reason to question whether the regulation of oocyte membrane ingression is important for polar body extrusion is that the KNL-1/BUB-1/CLS-2 sub-cortical patches are present only during meiosis I and are undetectable throughout all of meiosis II. While perhaps unlikely, undetectable levels of sub-cortical patches might suffice for such regulation during meiosis II. Further investigation to compare cortical actomyosin and microtubule distributions during meiosis I and II, and to assess the influence of eggshell synthesis and rigidity as meiosis proceeds, might reveal differences that preclude a need for microtubule regulation by sub-cortical patches during meiosis II.

Regardless of the relationship between oocyte membrane ingression and polar body extrusion, our results show that highly regulated microtubules are enriched in proximity to the plasma membrane, where they appear to modulate the extent of membrane ingression throughout the oocyte during meiosis I polar body extrusion.

## Limits to our analysis

While we hypothesize that the KNL-1/BUB-1/CLS-2 patches regulate sub-cortical microtubule dynamics, we cannot rule out the alternative possibility that spindle- and chromosome-associated, cytoplasmic, or other pools of these patch proteins influence sub-cortical microtubule dynamics. Optogenetic experiments designed to recruit sub-cortical patch components, or a heterologous microtubule destabilizing activity called Optokatanin [61], to limited areas of the oocyte cortex may provide more conclusive evidence that localized modification of microtubule dynamics can influence local membrane ingression independently of changes in cortical actomyosin dynamics.

Finally, a better understanding of oocyte cytoskeletal functions during polar body extrusion will require more comprehensive and higher resolution live cell imaging throughout all of meiosis I and II. Our spinning disk confocal microscopy does not provide sufficiently high spatial resolution to fully assess how sub-cortical microtubules and cortical actomyosin are distributed relative to each other. Recent advances in light microscopy include the use of near Total Internal Reflection Fluorescence microscopy to image the cell cortex in one-cell *C. elegans* zygotes at single molecule resolution [62]. Higher resolution imaging in oocytes will greatly improve our understanding of the functional relationships between the sub-cortical KNL-1/BUB-1/CLS-2 patches, sub-cortical microtubules, cortical actomyosin, and sites of membrane ingression during *C. elegans* polar body extrusion.

## Materials and methods

### *C. elegans* strain maintenance

All *C. elegans* strains used in this study are listed in S1 Table and were maintained at 20C on standard nematode growth medium plates seeded with *E. coli* strain OP50.

### Feeding RNAi Knockdown

All RNAi experiments were carried out by feeding *E. coli* strain HT115(DE3) induced by IPTG to express double-stranded RNA corresponding to the following genes: *bub-1*, *cls-2*, *klp-7*, *knl-*

*1*, *rod-1* and *zyg-9*. First bacteria clones were picked from an RNAi library and grown on LB Agar plate with Ampicillin [63]. Each bacterial clone was miniprepped using the Qiagen kit for subsequent sequence confirmation. Hypochlorite synchronized L1 larvae were grown on standard nematode growth medium plates, washed with M9 three times and then plated on the induced RNAi plates and grown at 20˚C until imaging. The feeding times were chosen such that if treatment were extended for an additional 6 more hours, 90% or more of the adult worms became sterile. For *bub-1*, *cls-2*, *klp-7*, *rod-1* and *zyg-9* worms were fed for 48–52 hours. For *knl-1* RNAi, L1 larvae were grown until L4 stage and placed onto *knl-1* RNAi plates for 35–40 hours. For auxin induced degron (AID) experiments, feeding times were chosen until hatching rate reached 0%. For all AID strains, hypochlorite synchronized L1 larvae were grown on standard nematode growth medium plates (NGM) until L4 stage, where worms were transferred onto NGM with 1mM Auxin plates. For all strains, worms were placed onto the 1mM Auxin plates for 13 hours before imaging.

## Nocodazole and taxol treatment

Young adult worms were dissected to release oocytes in 1ul egg salt buffer (118mM NaCl, 48mM KCl, 2mM CaCl2, 2mM MgCl2, and 0.025 mM of HEPES, filter sterilized before HEPES addition) containing 2% DMSO, 20 ug/ml nocodazole or 200nM taxol, or DMSO only for control. The coverslip with oocytes were placed over 3% agarose pad with egg salt buffer containing 1% DMSO, 10 ug/ml nocodazole in 1% DMSO, or 100 nM taxol in 1% DMSO, and then used immediately for imaging.

## Image acquisition

All imaging was carried out using a Leica DMi8 microscope outfitted with a spinning disk confocal unit–CSU-W1 (Yokogawa) with Borealis (Andor), dual iXon Ultra 897 (Andor) cameras, and a 100x HCX PL APO 1.4–0.70NA oil objective lens (Leica). (Molecular Devices) imaging software was used for controlling image acquisition. The 488nm and 561nm channels were imaged simultaneously with 1um Z-spacing.

*Ex utero* live imaging used the following parameters: 1μm Z spacing, 16 focal planes, 100ms exposure, 10s interval. Imaging was carried out by dissecting worms in 3ul egg salt buffer (118mM NaCl, 48mM KCl, 2mM CaCl2, 2mM MgCl2, and 0.025 mM of HEPES, filter sterilized before HEPES addition) on a coverslip before mounting onto a 2% agarose pad (diluted in egg salt buffer) on a microscope slide. Upon imaging for AID knockdown strains, worms were dissected onto egg salt buffer with 1 mM auxin and mounted onto a 2% agarose pad (diluted in egg salt buffer) with 1 mM auxin.

*In utero* live imaging used the following parameters: 1 μm Z spacing, 20 μm stacks, 80s exposure, 20s interval. Imaging was carried out by placing adult worms in 1.5 μl of M9 mixed with 1.5 μl of 1 μm polysterene microspheres (Polysciences Inc.) on a coverslip which was mounted carefully onto a 5% agarose pad.

## Image processing

General image processing was performed with ImageJ/Fiji software (National Institutes of Health). First, raw images were merged for the red/green channels then cropped to a 512x512 frame in order to make montages for each figure starting from Anaphase B to the end of anaphase B/end of meiosis I. Timing of each oocyte relied on the three-dimensional projection made by using Imaris software (Bitplane). Beginning of anaphase B was determined by the chromosome channel where the chromosomes were most compact during anaphase, as described elsewhere (REF). The end of anaphase B/meiosis I was the first timepoint when

oocyte chromosomes began to decondense after attempts at polar body extrusion [11]. For quantification analysis and figures, areas around oocytes were cropped out. Quantification of cortical microtubules and ingressions were analyzed in a normalized anaphase B time window to account for differing lengths of anaphase B duration in each genotype (S3A Fig). Oocytes were normalized by dividing the time intervals into four equal parts.

### Sub-cortical microtubule foci analysis pipeline

Sub-cortical microtubule foci (sCMF) quantification was carried out by initial processing using ImageJ/Fiji software followed by Imaris Software 9.8.2 (Bitplane). Macros available upon request.

Fiji/ImageJ was used to isolate locally strong TBB-2 signal and histone H2B signal.

The Fiji/ImageJ processing step produces a total of 5 channels: Channel 1 –Original Histone H2B (chromosome); Channel 2 –Original TBB-2 (microtubule); Channel 3 –H2B with Median, Gaussian, and Maximum Filters; Channel 4 –Tubulin with Remove Outliers, Subtraction and Median Filters; Channel 5 –Mask Segmentation (from the maximum intensity Z projection of Channel 2).

Once the raw image files for the GFP and mCherry channels were merged, the histone H2B and TBB-2 channels were filtered to isolate locally strong signals from each channel, and eliminate background. First, the original histone H2B channel was duplicated and processed with the following filters: Median filter with radius 5 pixels, Gaussian Blur, with sigma radius of 10 pixels, and Maximum filter with radius 5 pixels; refer to macro titled 'MedianGaussianBlur-Maximum.' These filters expand the isolated H2B signals to encompass the adjacent tubulin signals. This expansion was used to make surfaces in Imaris to eliminate sperm associated signals. This result were later be merged with the original merged image file as Channel 3.

Second, the TBB-2 channel was processed with the remove outliers Fiji function (Plugins/ Integral Image Filters/Remove Outliers) for background subtraction to isolate the sCMF signals visible in the GFP::TBB-2;mCherry::H2B associated strains with the following parameters: Block radius X 20 pixels, Block radius Y 20 pixels, and standard deviation .90 pixels. The result produced was subsequently used to model the background to subtract from the original TBB-2 channel using Fiji's Process/Image Calculator function. Furthermore, to remove small, noisy, or sparse signals the median filter was applied using a radius of 5 pixels. The final result produces an image of isolated, locally strong signal intensity, which includes sCMFs, sperm associated microtubules, spindle microtubules, and signals outside the oocyte. Refer to macro titled 'DuplicateAndRemoveOutlier.ijm' This result was later merged with the other created channels and original merged file as Channel 4.

Next, the original raw microtubule channel was duplicated again and a maximum intensity z-projection function was run to prepare for hand segmentation. For each timepoint, the region outside of the oocyte was hand segmented and cropped using Fiji's Edit/Clear Outside function. Afterward, the movie was converted to an 8-bit format. Refer to a macro titled 'Turn-To1and016bit.ijm.' Run the multiply function with a value of 255 followed by the divide function with a value of 255. Then convert back to 16-bit. This process converted the hand segmented picture into an 16-bit binary picture with values of 1 inside the ROI and values of 0 outside the ROI. This created a mask of the segmented oocyte so that Imaris processing (described below) could automatically make a surface roughly surrounding the oocyte without having to consider objects outside of the oocyte.

We then ran Fiji's Image/Stack/Tools/Concatenate function of this mask 4 times. This was to make a hyper stack with 16 slices in the z dimension from the mask derived from the Max projected image with only 1 z slice. Following this, run the function 'stack to hyperstack' with the following parameters: set to xyczt, channels– 1, slice– 16, and frames–number of frames in

**Table 1. Merge function for newly created channels.**

| Channel 1 | Histone H2B (Raw Image) |
| --- | --- |
| Channel 2 | Microtubules TBB-2 (Raw Image) |
| Channel 3 | Histone H2B with Median, Gaussian, and Maximum Filters |
| Channel 4 | Microtubules TBB-2 with Remove Outliers, Subtraction, and Median Filters |
| Channel 5 | Mask Segmentation |
| Channel 6 | Rectangle ROI |

To correct for 3D drift, channel 6 was used with the parameters shown in Table 2.

the original image file. The end product will be a mask of the segmented oocyte with 16 stacks in the z dimension. This result will later be merged as Channel 5.

Finally, refer to a macro titled 'MarkZandNextTimepoint.ijm'. This step was to mark the Z plane of the top surface of the oocyte. Duplicate the original TBB-2 channel. Starting from time point 1, draw a rectangle ROI around the oocyte of the outermost (cortical) z plane position of the oocyte. Run the macro for each timepoint, adjusting the z plane to the outermost cortical surface position that first shows the tubulin signal. This created a mask that has extremely high intensity values than the surrounding region and was automatically recognized by the 'Correct 3D Drift' plugin described below.

This result was later merged with the other created channels as Channel 6.

Run the merge function for all the newly created channels and the original Histone H2B and Microtubule TBB-2 channels as shown in Table 1.

## Imaris 9.8.2 (Bitplane)

**Surfaces 1—Oocyte.** Open the merged dataset containing the generated channels 1 through 5 in Imaris.

Create a new surface and refer to the custom creation parameter 'Oocyte.' Under algorithm settings mark track surfaces and object-object statistics, indicate source channel 5 (name not specified), smooth setting, thresholding absolute intensity, and surface detail defined at 0.294 um. Result should create a gray surface that encompasses the oocyte. Under the edit icon in the properties arena, mask all—channel 2 setting was used with the set voxel outside surface to 0. This created a masked channel 2.

**Table 2. Corrections for 3D drift.**

| Channel Registration | Channel 6 |
| --- | --- |
| Only consider pixels with values larger than 0 | Set to 0 |
| Lowest z plane to take into account | 1 |
| Highest z plane to take into account | 16 |
| Max shift x(pixels) | 10 |
| Max shift y(pixels) | 10 |
| Max shiftzy(pixels) | 90 |

This will correct drift based off of the channel 6 ROI. Among the resulting z-slices, slices with the oocyte signal are duplicated leaving z-slices outside the oocyte. At this time only channels 1–5 are duplicated because channel 6 will not be used hereafter. This image file with 5 channels will be used for Imaris (Bitplane) for further segmentation, drift correction and to make surfaces corresponding to the cortical microtubule signals, using the parameters summarized in Table 3.

**Table 3. Parameters for Imaris (Bitplane) image processing.**

| Parameter | yes/no | value |
|---|---|---|
| Segment only a region of interest | No | N/A |
| Track Surfaces (Over Time) | yes | N/A |
| Classify Surfaces | No | N/A |
| Object-Object Statistics | yes | N/A |
| Source Channel | N/A | 5 |
| Smooth (Surface Detail) [um] | yes | 0.294 |
| Thresholding (Absolute Intensity) | yes | Automatic |
| Filters | yes | 6.34 e4 voxels |
| Edit Surfaces | No | N/A |
| Tracking Algorithm | N/A | Autoregressive motion |
| Tracking Max Distance | N/A | 50um |
| Tracking Max Gap Size | yes | 0 |
| Fill gaps with all detected objects | No | N/A |
| Filter Tracks | No | N/A |

**Surfaces 2—Oocyte refine.** Deselect Surfaces 1, and select all channels except channel 1 and masked channel 2

Create a new surface and refer to the custom creation parameter 'Oocyte Refine.' This step was for further drift correction. The following settings were applied: Track surfaces, source channel 6 (also referred to as masked channel 2), threshold, algorithm autoregressive motion with parameters set to max distance 50 um and gap size 0. To correct drift, select all tracks and connected objects, correct image and all objects, translational and rotational drift, and include the entire result. Select channel 4 (the channel created in ImageJ) and masked channel 2. Under the edit icon indicate 'Mask All' for the channel 4 selection, with mask settings as constant inside/outside set voxels outside surface to 0, and applied to all time points. This results in a new masked channel 4.

**Surfaces 3—Nucleus expand.** Create new surface 3 and refer to the custom creation parameter 'NucleusExpand.' Set the source channel 3 with smoothing surface details .294 um and Absolute Intensity thresholding. Under the edit tab within properties arena set 'Mask All' with the following parameters: Channel Selection Channel 7—Masked Channel 4, Constant inside/outside set voxels inside surface to 0 um, and apply to all time points. This creates a "Masked Masked channel 4". This was used to eliminate the sperm associated TBB-2 signal from the isolated TBB-2 patch signals.

**Surfaces 4—Quantification of sCMFs.** Select Channel 1, 2, 4, and Masked Masked Channel 4 under the display adjustment. Create new surfaces 4 and refer to creation parameters titled 'CMPExUteroR2.' Select track surfaces, classify surfaces, and object-object statistics. Next select the source channel 8—Masked Masked Channel 4 with smoothing surfaces detail set to 0.294 um, and thresholding background subtraction Diameter of largest sphere which fits into the object 10 um. The result will show the filtered cortical microtubule surfaces. Manual deletion of surfaces outside of the oocyte were done for each time point. Set algorithm to autoregressive motion with max distance parameter 5 um and max gap size 3. From this point each of the surfaces will be defined and categorized by their mean intensity.

Deselect surfaces 4 and select surfaces 3. Under the edit tab select mask all and channel 6—masked channel 2. Under mask settings select constant inside/outside and set voxels outside surface to 0 um. This step was used to make a surface that encompasses the spindle. This creates a Masked Masked channel 2.

**Surfaces 5—Spindle.**   Deselect all display adjustment channels except for Channel 1 and Masked Masked channel 2. Create new surfaces 5 and choose spindle parameters with source channel 9—masked masked channel 2. Use the recommended threshold (absolute intensity) value. This creates a single surface of where the spindle was located through all time points. Furthermore, we ran the autoregressive motion algorithm with parameters set to max distance 50 um and max gap size 0.

From here the statistics for surfaces 4 that define sCMFs can be exported. sCMFs of which the distance from the spindle surface is larger than 0.1 um are subjected to statistical analysis. The voxel intensity cut off for detection of sCMFs surface was 150 of channel 4. For classification, sCMFs having mean intensity of channel 2 lower than 2303 were classified as weak, between 2303 and 2999 were classified as medium, higher than 2999 were classified as strong. These thresholds are set based on control movies, and were used for all other genotypes. See S13 Fig for histograms of three control movies with the cutoffs indicated.

## *C. elegans* global cortical ingression analysis pipeline

Manual segmentation was performed in FIJI to isolate dissected oocytes from adjacent dissected biological objects and subsequent automated segmentation and measurements were performed using python3 (skimage, scipy, tifffile, numpy, pandas, and matplotlib libraries). The algorithm developed as follows:

1. First, the oocyte cortex was modeled as a contour (S14A Fig).

2. To measure the length of the ingressions, the convex hull of the cortex contour was used to establish a measurement reference to the ingressions. Given a set of points the convex hull was the subset of points that form lines which, when connected, enclose all other points (S14B Fig).

3. In the absence of ingressions, the contour and convex hull completely overlap; when ingression begins, the convex hull reference lines emerge from the ingressing contour (S14C Fig).

4. Measurements were made piece-wise between adjacent intersection points of the convex hull and contour. The ingression measurement could then be transformed from 2D coordinate space to 1D distance space by finding the distance from the contour at any point to the reference line of the convex hull. Python's scipy library was used to find peaks in the distance space and any peak distance longer than $0.441\mu$m (3 pixels) from the convex hull was recorded as an ingress length. Distances were calculated using the following equation. $A$ and $C$ are the adjacent intersection points between the convex hull and contour, $B$ was any point along the contour between $A$ and $C$ (S14D Fig). These distances are calculated by using the length of the cross product between vectors $\overrightarrow{AB}$ and $\overrightarrow{AC}$, normalized by the length of vector $\overrightarrow{AC}$

$$d \quad = \frac{\| \overrightarrow{AB} \times \overrightarrow{AC} \|}{\| \overrightarrow{AC} \|}$$

## Algorithm

1. Movies were first hand-segmented using Fiji's elliptical brush tool to remove adjacent oocytes and non-relevant biological material.

2. A median filter using a 2×2 pixel kernel was performed on each frame followed by intensity-based threshold using Yen's algorithm to produce a binary image of the oocyte cortex [64].

3. When the contour ingresses and appears to form a line (narrow, deep ingress), the contour model fails to fully capture the length of the ingression. To account for this, the following additional steps were added to the segmentation process to minimally accentuate the morphology in a uniform way (S14D Fig).

   a. Once the image was binarized after Yen thresholding, it was then dilated by 1 pixel and inverted.

   b. The inverted image has now two non-zero objects, the interior of the cortex and the exterior which was shaped by the image edges.

   c. The exterior, non-zero object touching the image edge was removed leaving the interior as the basis for the contour modeling. This interior retains the extent of these narrow, deep ingressions.

Once the images are fully segmented, the convex hull and contour are calculated and their intersection points found. These points serve as the basis of the analysis above.

Ingressions within close proximity to chromosome location (also referred to as spindle-associated ingressions) were excluded in the analysis. To discount these ingressions, the maximum intensity projection of the chromosome channel was used as a positional mask against the cortex channel.

   d. Yen thresholding was used on the maximum intensity projection of the chromosome channel along the focal axis (z).

   e. The pixel locations of the binary chromosome objects were stored for every time point.

   f. Ingress peak positions found to be within $3.23\mu$m (22 pixels) of the chromosome objects were classified as being spindle-associated ingressions and not included into the final analysis. The constraint of 22 pixels was measured to be the largest possible search radius from chromosome edges to potential ingress edges to exclude main ingressions that included multiple ingression points in their entirety.

Ingression lengths were exported into an excel sheet. Ingression lengths are filtered out for any ingressions greater than 1 micron and any ingressions in close proximity to chromosomes. Mean length, mean sum, and mean number are calculated in the normalized anaphase B time window.

### Sub-cortical patch quantification

Sub-cortical ROD-1/KNL-1/BUB-1/CLS-2 patch quantification was carried out in ImageJ/Fiji software. Timing of each oocyte was determined by anaphase B chromosome compaction. For all oocytes in each condition, 5 timepoints were taken for quantification. The timepoints were determined as follows: 1) Anaphase B chromosome compaction, set at 0s 2) -200s prior to anaphase B chromosome compaction. The sum of the first 5 timepoints starting from -200s (-200s, -190s, -180s, -170s prior to beginning of anaphase B) were used to compare the total raw integrated pixel intensity of the foci across all conditions. Next, a max intensity z projection was made from the raw image. On the max intensity projected image, a median filter with radius 3, and background subtraction with rolling = 20 was used. The oocyte was hand segmented at each timepoint and added to the ROI manager. From there, we used the Triangle Thresholding Algorithm (https://bioimagebook.github.io/CHANGELOG.html) onto the ROI and "Analyze Particles" to acquire the raw integrated pixel intensity for all foci. To only measure cortical foci, we subtracted the chromosome-associated signal from the measurements.

To measure foci depletion, we first took the sum of raw integrated pixel intensities for each oocyte. Next, we took the mean of the sum of raw integrated pixel intensities for each condition and normalized those values on a 0 to 100 scale based off of minimum value set to 0 and maximum value set to the maximum value across all conditions within each GFP fusion (CLS-2::GFP, BUB-1::GFP, KNL-1::GFP). Normalization formula was as follows: $z_i = (x_i - min(x)) / (max(x) - min(x)) * 100$ where $z_i$ is the normalized value in the dataset, $x_i$: the value that needs to be normalized in the dataset, $min(x)$: the minimum value set to 0, and $max(x)$: the maximum value in the dataset. For calculating percent depletion, 100 was subtracted from the normalized values ($z_i$).

## Quantification of microtubule levels after nocodazole and taxol treatment

The same procedure described above for sub-cortical patch quantification was used to quantify microtubule signal intensity for nocodazole and taxol-treated oocytes, except we used the sum of 16 time points for quantification of taxol-treated oocytes, which exhibit more variability in microtubule levels compared to nocodazole-treated oocytes.

## Statistics

P-values comparing distributions for microtubule and ingression quantification were calculated using the Mann-Whitney U-test. Statistical analysis was performed using Microsoft Excel (Microsoft) and Prism 9 (GraphPad Software) and graphs were made in Prism 9 (see S1–S5 Files).

## Supporting information

**S1 Fig. CLS-1, BUB-1 and KNL-1 sub-cortical patches are present only during meiosis I, not during meiosis II.** Maximum intensity projections of all focal planes during meiosis I in live ex utero oocytes expressing mCherry::H2B (magenta) to mark chromosomes (lower rows) and GFP (cyan) fusions (upper rows) to CLS-2 (A), BUB-1 (B), and KNL-1 (C). All three proteins were associated with the oocyte spindle and chromosomes, and were present in sub-cortical patches, during meiosis I, but were associated only with the spindle and chromosomes, and not present in sub-cortical patches, during meiosis II.
(PDF)

**S2 Fig. Quantification of sub-cortical patch intensity after RNAi knockdowns.** (A) Maximum intensity projections (MIPs) of the five surface-most focal planes (upper row), and the same MIPs after a Triangle threshold filter to identify sub-cortical patches (lower row) in an *ex utero* oocyte expressing a CLS-2::GFP fusion, over 5 time points preceding the start of anaphase B. (B) Quantification of threshold-limited sub-cortical patch intensities, after manually removing spindle associated signal, in control and mutant *ex utero* oocytes expressing GFP fusions to CLS-2, BUB-1 and KNL-1 as indicated, showing scatter plots for raw integrated pixel intensity and bar graphs for per cent depletions. See S1 File for raw data on patch quantification.
(PDF)

**S3 Fig. Anaphase B membrane ingression in control and mutant oocytes.** Time projections of single central focal planes throughout meiosis I anaphase B from six *ex utero* oocytes of each genotype or condition, all expressing GFP::PH (cyan) and mCherry::H2B (not shown) to mark oocyte membranes and chromosomes, respectively.
(PDF)

**S4 Fig. Cell division defects and computational quantification of membrane ingression during *C. elegans* oocyte meiosis I.** (A) Anaphase B duration in oocytes of each genotype or condition, with mean and standard deviation indicated. Mean in minutes are shown above each scatter plot. See S5 File for raw data. (B) Bar graphs indicating number of embryos scored and percent of each genotype or condition in which polar body extrusion after meiosis I was successful, with some mCherry::H2B signal detected in an intact polar outside of the oocyte after the completion of meiosis I. See S5 File for raw data (C) Schematic of a convex hull and contour model to measure furrow lengths in *C. elegans* oocytes. Defects in the contour are shown in middle oocyte and are quantified using a distance formula (see Materials and Methods). (D) Single focal planes of live control and mutant oocytes strains that express GFP::PH (white) and mCherry::H2B (not shown) processed for scoring ingression. Pink dots indicate furrows that are computationally counted and measured in length (see Materials and Methods). (E) Relative standard deviation of five surface-most focal planes ten seconds after start of anaphase B in *ex utero* control and *cls-2* mutant oocytes. Maximum intensity projections images of surface-most five focal planes from *ex utero* oocytes 10 seconds after anaphase B onset were generated, excluding a circular region with an 80 (11.75 µm) pixel radius surround the spindle were hand segmented and subjected for quantification of mean pixel intensity and pixel intensity standard deviation. Relative standard deviation was calculated by dividing standard deviation by the mean. See S4 File for raw data.
(PDF)

**S5 Fig. ROD-1 is required for KNL-1, BUB-1 and CLS-2 to localize to sub-cortical patches.** (A) Maximum intensity projections (MIPs) of the five surface-most focal planes, merged with 5 internal focal planes that include most of the chromosomes, in live control (upper row) or *rod-1(RNAi)* (lower row) *ex utero* oocytes expressing mCherry::H2B to mark chromosomes (magenta) and a GFP fusion to ROD-1, CLS-2, BUB-1 or KNL-1 (cyan). (B) Selected and merged focal planes from *ex utero* oocytes during meiosis I anaphase B that express GFP::PH (cyan) and mCherry::H2B (magenta) to mark the plasma membrane and chromosomes, respectively. A single central focal plane is shown for the membrane, merged with a maximum intensity projection of 5 consecutive internal focal planes that encompass most of the oocyte chromosomes. (C) Quantification of the mean length of membrane ingressions that were 1µm or more in length, the mean sum of lengths, and the mean number of membrane ingressions, over normalized anaphase B time in control and *rod-1(RNAi)* oocytes. The mean length and mean sum of lengths, but not the number, of ingressions were significantly increased in *rod-1* mutant oocytes. (D) Quantification of threshold-limited sub-cortical patch intensities in control and *rod-1(RNAi)* oocytes expressing GFP fused to ROD-1 and mCherry::H2B, after manually removing spindle associated signal, showing scatter plots for raw integrated pixel intensities and bar graphs for per cent depletion.
(PDF)

**S6 Fig. Imaris quantification of sub-cortical microtubule patches (sCMFs) in control and *cls-2(or1948)* oocytes.** (A) Merged maximum intensity projections of *ex utero* control and *cls-2(or1948)* oocytes that express GFP::TBB-2 (white) to mark microtubules, and mCherry::H2B to mark chromosomes (not shown). Top rows for control and *cls-2(or1948)* oocytes show GFP::TBB-2 (white); middle rows show color-coded Imaris surfaces, based on our cutoffs for weak (blue), medium (purple), and strong (red) sCMFs (see Materials and Methods); bottom rows show merges of GFP::TBB-2 and Imaris surfaces. (B) Quantification of weak, medium and strong sCMFs in single control and *cls-2(or1948)* oocytes during anaphase B. The *cls-2 (or1948)* oocyte shown has mostly weak sCMFs, with a loss of medium and strong sCMFs

throughout anaphase B.
(PDF)

**S7 Fig. Taxol treatment suppresses excess membrane ingression in *cls-2(or1948)* oocytes.**
(A) Selected and merged focal planes from live control (DMSO treated; upper row) and taxol-treated (three lower rows) *ex utero cls-2(or1948)* oocytes, during meiosis I anaphase B, that express GFP::PH (cyan) and mCherry::H2B (magenta) to mark the plasma membrane and chromosomes, respectively. A single central focal plane is shown for the membrane, merged with a maximum intensity projection of 5 consecutive internal focal planes that encompass most of the oocyte chromosomes. (B) Quantification of the mean length of membrane ingressions that were 1µm or more in length, the mean sum of lengths, and the mean number of membrane ingressions, over normalized anaphase B time in control and taxol-treated oocytes. The mean length, mean sum of lengths, and mean number of ingressions were all significantly decreased in taxol-treated *cls-2(or1948)* oocytes. (C) Five surface-most focal planes (left column) or five central focal planes (middle column), merged with five internal focal planes that include most of the chromosomes, from live *ex utero* control, nocodazole-treated and taxol-treated oocytes expressing GFP::TBB-2 and mCherry::H2B to mark microtubules and chromosomes, at ¾ of normalized time through anaphase B. Surface (black) and central (red) line scans (right column) are from vertical magenta lines in oocyte images. Scale bar = 10 µm. (D) Quantification of microtubule levels in control, nocodazole-treated and taxol-treated oocytes. See S1 File for raw data on microtubule levels comparisons.
(PDF)

**S8 Fig. Cortical non-muscle myosin distribution in control oocytes.** Merged maximum intensity projections (MIPs) during meiosis I anaphase B of five *ex utero* control oocytes expressing NMY-2::GFP (cyan) and mCherry::H2B (magenta) to mark NMY-2/non-muscle myosin II and chromosomes, respectively. MIPs of five surface-most focal planes showing non-muscle myosin are merged with MIPs of five consecutive internal focal planes that encompass most of the oocyte chromosomes, visible at the left, anterior end of each oocyte. A roughly even but variable and dynamic network of cortical NMY-2::GFP foci persists throughout most of anaphase B until dissipating near the end.
(PDF)

**S9 Fig. Cortical non-muscle myosin distribution in *cls-2* mutant oocytes.** Merged maximum intensity projections (MIPs) during meiosis I anaphase B of five *ex utero cls-2(or1948)* oocytes expressing NMY-2::GFP (cyan) and mCherry::H2B (magenta) to mark NMY-2/non-muscle myosin II and chromosomes, respectively. MIPs of five surface-most focal planes showing non-muscle myosin are merged with MIPs of five consecutive internal focal planes that encompass most of the oocyte chromosomes, visible at the left, anterior end of each oocyte. Abnormal linear arrays of NMY-2::GFP foci are present in some *cls-2* mutant oocytes (arrowheads).
(PDF)

**S10 Fig. Cortical non-muscle myosin distribution after KLP-7 knockdown.** Merged maximum intensity projections (MIPs) during meiosis I anaphase B of five *ex utero klp-7(RNAi)* oocytes expressing NMY-2::GFP (cyan) and mCherry::H2B (magenta) to mark NMY-2/non-muscle myosin II and chromosomes, respectively. MIPs of five surface-most focal planes showing non-muscle myosin are merged with MIPs of five consecutive focal planes that encompass most of the oocyte chromosomes,visible at the left, anterior end of each oocyte.
(PDF)

**S11 Fig. Cortical non-muscle myosin distribution after ZYG-9 knockdown.** Merged maximum intensity projections (MIPs) during meiosis I anaphase B of five *ex utero zyg-9(RNAi)* oocytes expressing NMY-2::GFP (cyan) and mCherry::H2B (magenta) to mark NMY-2/nonmuscle myosin II and chromosomes, respectively. MIPs of five surface-most focal planes showing non-muscle myosin are merged with MIPs of five consecutive internal focal planes that encompass most of the oocyte chromosomes, which are visible at the left, anterior end of each oocyte.
(PDF)

**S12 Fig. Contractile ring dynamics during meiosis I polar body extrusion are defective in control and mutant oocytes.** Projections of all focal planes after Imaris-mediated rotation to obtain ring-centric, more oocyte-end-on views of contractile ring dynamics in four each of control and mutant oocytes expressing GFP::NMY-2 (cyan) and mCherry:H2B (magenta) to mark NMY-2/non-muscle myosin II and chromosomes, respectively, in control and mutant oocytes (see Materials and Methods).
(PDF)

**S13 Fig. Histograms of sub-cortical microtubule foci in three control oocytes.** Histograms are based on the mean intensity of all surfaces. Each surface represents a single sCMF. The histograms are automatically made in Imaris throughout the course of image processing. The classification threshold between class for weak and medium 2303 was set to separate the left most mountain, best represented in Control Oocyte 1 and Control Oocyte 2. Control oocyte 1 was a darker example of the three presented, and it shows a high frequency of weak sCMFs. Control oocyte 2 was a brighter example, and the thresholding cutoffs are represented with the thresholding covering the whole mountain within the weak sCMF range, and lower frequency for medium, and strong sCMFs. The weak class does not cover the whole left-most mountain in control oocyte 3, a very bright example, and sCMFs with high intensity in the mountain are classified as medium. But the bright foci outside the mountain are classified as the strong class with the threshold 2999.
(PDF)

**S14 Fig. Development of oocyte membrane ingression pipeline algorithm.** (A) Contour model of an oocyte cortex without ingressions. (B) Visual description of a convex hull. (C) Contour and convex hull model of the oocyte cortex without (left) and with (right) ingressions. (D) Schematic for the calculation of ingression lengths from the line joined by intersection points A and C to any point B along the contour. (E) Inverting the intensity-based segmentation of the cortex allows for a better contour model of the ingressions. The contour on the bottom left uses the intensity-based threshold of the cortex fluorescence as the input for the contour model (exterior contour model), but poorly captures the narrow ingresions. Instead we used the cortex intensity as a border and the inverted interior as the input for the contour model (bottom right).
(PDF)

**S1 Movie. KNL-1/BUB-1/CLS-2 cortical patch dynamics during meiosis I and II.** Full maximum intensity projections of all focal planes from (top row) *in utero* oocytes from nuclear envelope breakdown to end of meiosis I, and (bottom row) *ex utero* oocytes from meiosis I metaphase to end of meiosis II.
(AVI)

**S2 Movie. Co-localization of KNL-1, BUB-1 and CLS-2 during meiosis I.** Merged maximum intensity projections of five surface-most focal planes and five consecutive internal focal planes

that encompass most of the oocyte chromosomes.
(AVI)

**S3 Movie. Membrane dynamics in control and mutant oocytes during meiosis I anaphase B.** Merged single central focal plan for membrane and five consecutive focal planes that encompass most of the oocyte chromosomes.
(AVI)

**S4 Movie. Cortical microtubules in control and mutant oocytes during meiosis I anaphase B.** Merged maximum intensity projections of five surface-most focal planes and five consecutive internal focal planes that encompass most of the oocyte chromosomes.
(AVI)

**S5 Movie. Imaris spots quantification of microtubules in control and *cls-2* mutant oocytes.** Projections of all focal planes after Imaris-mediated rotation.
(AVI)

**S6 Movie. KNL-1 sub-cortical patch dynamics after nocodazole treatment.** Merged maximum intensity projections of five surface-most focal planes and five consecutive internal focal planes that encompass most of the oocyte chromosomes.
(AVI)

**S7 Movie. Cortical NMY-2/non-muscle myosin distribution in control and mutant oocytes during meiosis I anaphase B.** Merged maximum intensity projections of five surface-most focal planes and five consecutive internal focal planes that encompass most of the oocyte chromosomes.
(AVI)

**S8 Movie. Imaris-rotated views of contractile ring dynamics in control and mutant embryos during meiosis I anaphase B.** Projections of all focal planes after Imaris-mediated rotation.
(AVI)

**S1 Table. *C. elegans* strains used in this study.**
(PDF)

**S1 File. Raw data for calculation of sub-cortical patch quantification and microtubule levels after nocodazole and taxol treatment.**
(XLSX)

**S2 File. Raw data for calculation of membrane ingression p values.**
(XLSX)

**S3 File. Raw data for linescans.**
(XLSX)

**S4 File. Raw data for relative standard deviation comparison.**
(XLSX)

**S5 File. Raw data for calculation of sub-cortical microtubule foci p values.**
(XLSX)

**S6 File. Phylogenetic diversity in oocyte meiotic cell division mechanisms.**
(DOCX)

## Acknowledgments

We thank Arshad Desai, Julien Dumont, Reto Gassmann, Frank McNally, and the *Caenorhabditis* Genetics Center (funded by the NIH Office of Research Infrastructure Programs) for *C. elegans* strains, Dan Dickinson and Alexander Cartagena-Rivera for helpful discussions, and Chris Doe for sharing equipment.

## Author Contributions

**Conceptualization:** Alyssa R. Quiogue, Adam Fries, Bruce Bowerman.

**Data curation:** Alyssa R. Quiogue, Eisuke Sumiyoshi.

**Formal analysis:** Alyssa R. Quiogue, Eisuke Sumiyoshi, Adam Fries, Bruce Bowerman.

**Funding acquisition:** Bruce Bowerman.

**Investigation:** Alyssa R. Quiogue, Eisuke Sumiyoshi, Adam Fries, Chien-Hui Chuang, Bruce Bowerman.

**Methodology:** Alyssa R. Quiogue, Eisuke Sumiyoshi, Adam Fries, Chien-Hui Chuang, Bruce Bowerman.

**Project administration:** Bruce Bowerman.

**Resources:** Bruce Bowerman.

**Software:** Eisuke Sumiyoshi, Adam Fries.

**Supervision:** Bruce Bowerman.

**Validation:** Alyssa R. Quiogue, Eisuke Sumiyoshi, Adam Fries, Chien-Hui Chuang, Bruce Bowerman.

**Visualization:** Alyssa R. Quiogue, Eisuke Sumiyoshi, Adam Fries, Bruce Bowerman.

**Writing – original draft:** Alyssa R. Quiogue, Bruce Bowerman.

**Writing – review & editing:** Alyssa R. Quiogue, Eisuke Sumiyoshi, Adam Fries, Bruce Bowerman.

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
