## [Decision Letter · Decision Letter 0]

14 Jun 2023

Dear Dr Bowerman,

Thank you very much for submitting your Research Article entitled 'Cortical microtubules oppose actomyosin-driven membrane ingression during C. elegans meiosis I polar body extrusion' to PLOS Genetics.

The manuscript was fully evaluated at the editorial level and by two independent peer reviewers. The reviewers appreciated the attention to an important topic but identified some concerns that we ask you address in a revised manuscript.

We therefore ask you to modify the manuscript according to the review recommendations. Your revisions should address the specific points made by each reviewer. This will include: 1) making changes and clarifications to the text to ensure alternative models and interpretations are presented; 2) improving some of the figures (i.e., ensuring these are representative and clear images); 3) additional experiments with particular attention to the three main experiments suggested by Reviewer 1; 4) providing evidence of degree of depletion observed upon RNAi; and 5) providing missing quantification (please see comments by both reviewers about this).

Yours sincerely,

Monica P. Colaiácovo

Academic Editor

PLOS Genetics

Gregory P. Copenhaver

Editor-in-Chief

PLOS Genetics

Reviewer's Responses to Questions

**Comments to the Authors:**

Reviewer #1: Polar body extrusion during female meiosis is required to produce zygotes with a correct chromosome number in nearly all animals and is thus a critically important process that requires coordination between spindle microtubules and cortical actomyosin. In this manuscript, the authors show that microtubules near the cortex far from the spindle inhibit myosin-dependent invaginations of the cortex far from the spindle. The authors speculate that their findings indicate something about how polar body extrusion at the spindle works, but this connection is not completely clear. The quality of the time-lapse imaging data, image analysis, and genetic manipulation is very high. With 3 experiments and several writing adjustments, this manuscript would be very appropriate for publication in PLoS Genetics.

Required experiments:

1.) In the introduction and results, CLS-2 is described as localizing in cortical patches that are equated with linear elements. Linear elements are ROD-1-dependent and neither polar body defects nor ectopic plasma membrane ingressions have been reported after ROD-1 depletion. The authors should test whether the cortical patches reported in Fig. 2 are dependent on ROD-1. If they are dependent on ROD-1, the authors should report on membrane invaginations in rod-1-depletions. If they are independent of rod-1, this should be clarified in the text and they should not be called linear elements. The authors should also show or cite data showing a lack of colocalization of patches with myosin, actin, and a GTP-rho biosensor (because these all localize in cortical patches). Because patches are not present during meiosis II, the authors should explain how they could be required for polar body extrusion.

2.) The data on cortical microtubule patches (CMPs) presented in Figure 5 needs some improvement in presentation because these patches are completely not obvious from the images shown. Do these cortical microtubule patches co-localize with CLS-2/BUB-1/KNL-1 cortical patches? If the same image analysis is applied to subcortical regions, are there cortical patches in the cytoplasm (not at the cortex)? What do these authors hypothesize these cortical microtubule patches represent?

3.) Excessively deep membrane invaginations in knl-1(RNAi) or bub-1(RNAi) were indistinguishable from those in cls-2(RNAi) but polar body extrusion occurred in 80% of embryos. It is suggested that this is due to partial knockdown by RNAi, however, the simple interpretation is that limiting membrane invaginations through BUB1/KNL-1 is not required for polar body extrusion. If the authors wish to make an argument based on partial depletion, they should show the extent of depletion. Either the authors need to achieve a higher degree of knockdown, or the overall significance of regulating invaginations over the entire embryo requires more careful explanation. Documentation of the extent of knockdown of csnk-1 is also essential for the conclusion made in the discussion based on a lack of phenotype for csnk-1(RNAi).

Major improvements to writing:

In the introduction, the authors should bring up the two possible mechanisms by which microtubules might inhibit actomyosin-dependent invagination of the plasma membrane, direct coupling by mechanical attachment of microtubules to the plasma membrane or inhibition of actomyosin by microtubules. It is well known that astral microtubules emanating from centrosomes regulate cortical actomyosin but this is not mentioned until Fig. 9 of the results and is inappropriately discounted in the discussion. This more unbiased writing is especially important since no data directly measuring cortical stiffness is presented and the authors never clearly state how microtubules would generate stiffness. The possible role of osmotic pressure in stiffness should also be brought up in a more unbiased manner.

Throughout the manuscript, the use of the words “stiffness” and “elasticity” and the apparent assumption of an inverse relationship between them is confusing and somewhat over-stated. Neither physical parameter is measured experimentally here (for example by needle deflection experiments that have been published for mouse oocytes). Stiffness has units of N/m. The inverse of stiffness, compliance, has units of m/N. Elasticity has units of N/m2. Several sentences in the manuscript do not make sense. This reviewer understands how stiffness could resist deformation (invagination) by myosin contractility. This reviewer does not understand the assumed relationship with elasticity. It is up to the authors to explain this relationship.

This manuscript addresses the function of “cortical microtubules” implying that there is a subset of microtubules that are in some way attached to the cortex. Because microtubules fill the entire volume of the meiotic embryo and there is no evidence of a mechanical attachment between microtubules and the plasma membrane and typical minus-end binding proteins do not localize at the cortex, the authors need to more carefully describe what is known about the polarity of cortical or cytoplasmic microtubules. Cortical microtubules have been characterized in Drosophila oocytes and epithelial cells of C. elegans and vertebrates. In all of these cases, g-tubulin is concentrated at the cortex implying attachment of microtubule minus ends to the cortex. G-tubulin does not localize to the cortex of C. elegans meiotic embryos making the polarity and mechanical attachment unclear. Minus ends at the cortex have been suggested by the phenotype of kinesin-1 mutants by McNally 2010 Dev Bio 339:126 and Kimura 2017 Nat Cell Bio 19:399.

Minor improvements to writing:

The subheading “KNL-1, BUB-1 and CLS-2 stabilize cortical microtubules during meiosis I polar body extrusion” should be changed to “KNL-1, BUB-1 and CLS-2 increase the number of cortical microtubule patches during meiosis I polar

body extrusion” since no measure of microtubule stability is shown.

No mutant that specifically eliminates CLS-2/BUB-1/KNL-1 patches is described making it difficult to assess the significance of these patches.

Fig. 3B needs quantification and more explanation. How many embryos displayed the phenotype shown? Measurements of fluorescence intensity showing the variability would be more rigorous than just stating a black and white dependence. This analysis should also show the extent of depletion of the protein targeted by RNAi. It is not possible to conclude that KNL-1 localization is independent of CLS-2 unless complete loss of CLS-2 is demonstrated. The legend states that the images shown were acquired at “roughly metaphase”. A brief explanation of how metaphase was scored (especially in the knl-1 depletion) would make the result more reproducible by others.

Adding page and line numbers to the manuscript would make life easier for a reviewer.

In most of the manuscript, references are called out with a number while in some locations references are listed by author name and year.

Some of the writing needs to be condensed (eg. the legends for Fig. 1 and 4, and the discussion).

The authors should at least comment on whether the plasma membrane marker (PH domain of rat phospholipase C zeta which binds specifically to PIP2) could be influencing their results. At a minimum, a description of the properties of this marker should be included.

How is the onset of anaphase B determined in strains expressing mCherry::H2b and GFP::PH when distinguishing between anaphase A and anaphase B requires imaging the spindle poles? The results text states that maximum chromatin compaction was used for cls-2-depleted embryos but the legend implies this was used for all genotypes. This should be clarified.

“Taxol treatment substantially elevated cortical microtubule levels during meiosis I (Fig 6A, Movie 3)” should show some quantitation rather than just a single image.

Polar body extrusion appears normal without membrane invaginations in taxol-treated oocytes, again raising the question of significance of the invaginations around the entire embryo.

Fig. 7G right 2 panels have a comparison of 3 data sets but only one significance. This needs to be clarified especially for the far right. The conclusion in the text is that cls-2; klp-7 did not have a significant effect but the figure shows p<.01. This needs to be clarified.

The lack of a phenotype after csnk-1(RNAi) used as evidence that hydrostatic pressure is not involved in polar body extrusion is beyond mystifying.

Reviewer #2: Quiogue et al address the role of non-chromosome associated outer kinetochore protein assemblies on membrane ingression during C. elegans meiosis I. This is a very interesting subject because since their identification decades ago, these patches, rods, or linear elements, as they have been referred to, their function has not been addressed. The authors of this manuscript focus on a subset of these proteins, the outer kinetochore scaffold, KNL-1, the kinase BUB-1, and the microtubule stabilising protein CLS-2.

The authors show that KNL-1, BUB-1, and CLS-2 are required to limit membrane ingression during meiosis I. Ultimately, the authors suggest that CLS-2 as part of these non-chromosomal outer kinetochore patches, stabilises microtubules and hence stiffen the oocyte cortex, limiting membrane ingression.

Even though it is not clear how these patches could regulate microtubule stability (which do not co-localise with cortical microtubules), the observations are interesting and are likely to contribute to further investigate and understand the role of these elusive protein assemblies. The experiments are well performed and thoroughly analysed. Furthermore, there is ample explanation on the quantitative analysis. I have some questions/comments that the authors might want to look into, but I am overall in favour of publication.

- The key big question is how the authors envision that these patches regulate cortical MTs and what would be the ‘anchor’ for these patches, since presumably CENP-A and CENP-C are not present. Is MEL-28 required for KNL-1?

- The authors mention in the introduction that CLS-2 is part of a kinetochore sub-complex that includes BUB-1 and HCP-1/2. While BUB-1 can interact with HCP-1/2 and the latter with CLS-2, we don’t know whether this is a subcomplex and if so, its composition. Additionally, the co-localisation of BUB-1, HCP-1/2, and CLS-2 is highly dynamic, and it is lost as anaphase progresses. Therefore, pointing to these specific proteins as a complex might give the reader the wrong impression that these proteins interact throughout meiosis I.

- Similar point as above, when the authors mention that BUB-1 and CLS-2 but not KNL-1 localise to central spindle, this is not the full picture, since the two proteins occupy different regions within the segregating chromosomes and are therefore likely not forming a complex at this stage. They might be interacting in the patches, which might show different dynamics than spindle or chromosome-associated kinetochore proteins.

- In Figure 3B, care should be taken when assessing the dependencies for each protein localisation since this is being analysed with transgenes (in the background of untagged endogenous protein) and their levels might differ significantly from the endogenous proteins. Maybe using endogenously tagged proteins and/or immunofluorescence would be important here. Additionally, the fact that depletions are partial complicates the analysis. And lastly, it appears as if bub-1(RNAi) reduces the KNL-1 cortical signal but the authors say the opposite. Some form of quantitation would make it easier for the reader to follow.

- Still in Figure 3, do KNL-1 and BUB-1 localise normally in cls-2 mutant? This is important because the authors switch from RNAi to the mutant at this point in the paper.

- In Figure 4, there seems to be significant ingression in or around the spindle area in bub-1(RNAi) and knl-1(RNAi) oocytes, but not in cls-2 mutant oocytes, where these ingressions appear outside the spindle region. Additionally, while the image shown for knl-1(RNAi) shows a great degree of ingression in the anterior region, the quantitation shows no significant different vs control. Is this not a representative image?

- In Figure 6, does nocodazole treatment affect KNL-1, BUB-1, or CLS-2 localisation? Does taxol treatment rescue the defect upon bub-1(RNAi), knl-1(RNAi), or cls-2(or2948)?

- The authors point to the RZZ complex in the discussion (without mentioning it specifically). Given the ability of the RZZ complex to oligomerise, could the RZZ be the ‘scaffold’ of these patches? Have the authors tested RZZ components localisation and/or the effect of their depletion on cortical MTs?

- Check citations format. Some appear as numbers, some are cited by name, and also the text ‘(REF)’ appears.

**Have all data underlying the figures and results presented in the manuscript been provided?**

Reviewer #1: Yes

Reviewer #2: Yes

PLOS authors have the option to publish the peer review history of their article (what does this mean?). If published, this will include your full peer review and any attached files.

Reviewer #1: No

Reviewer #2: **Yes: **Fede Pelisch

---

## [Editor Report · Decision Letter 1]

19 Sep 2023

Dear Dr Bowerman,

We are pleased to inform you that your manuscript entitled "Microtubules oppose cortical actomyosin-driven membrane ingression during C. elegans meiosis I polar body extrusion" has been editorially accepted for publication in PLOS Genetics. Congratulations!

Yours sincerely,

Monica P. Colaiácovo

Academic Editor

PLOS Genetics

Gregory P. Copenhaver

Editor-in-Chief

PLOS Genetics

Comments from the reviewers (if applicable):

**Data Deposition**

http://datadryad.org/submit?journalID=pgenetics&manu=PGENETICS-D-23-00571R1

**Press Queries**

---

## [Editor Report · Acceptance letter]

29 Sep 2023

PGENETICS-D-23-00571R1 

Microtubules oppose cortical actomyosin-driven membrane ingression during C. elegans meiosis I polar body extrusion 

Dear Dr Bowerman, 

We are pleased to inform you that your manuscript entitled "Microtubules oppose cortical actomyosin-driven membrane ingression during C. elegans meiosis I polar body extrusion" has been formally accepted for publication in PLOS Genetics! Your manuscript is now with our production department and you will be notified of the publication date in due course.

With kind regards,

Anita Estes

PLOS Genetics

On behalf of:
